# Apoptosis-mediated ADAM10 activation removes a mucin barrier promoting T cell efferocytosis

Linnea Z. Drexhage [1,8], Shengpan Zhang [1,8], Maeva Dupont[1,2], Franziska Ragaller [3], Ellen Sjule [3], Jose Cabezas-Caballero [1], Lachlan P. Deimel [1], Helen Robertson [1], Rebecca A. Russell [1,4], Omer Dushek [1], Erdinc Sezgin [3], Niloofar Karaji[1,5] ✉ & Quentin J. Sattentau [1,6,7] ✉

Efferocytic clearance of apoptotic cells in general, and T cells in particular, is required for tissue and immune homeostasis. Transmembrane mucins are extended glycoproteins highly expressed in the cell glycocalyx that function as a barrier to phagocytosis. Whether and how mucins may be regulated during cell death to facilitate efferocytic corpse clearance is not well understood. Here we show that normal and transformed human T cells express a subset of mucins which are rapidly and selectively removed from the cell surface during apoptosis. This process is mediated by the ADAM10 sheddase, the activity of which is associated with XKR8-catalyzed flipping of phosphatidylserine to the outer leaflet of the plasma membrane. Mucin clearance enhances uptake of apoptotic T cells by macrophages, confirming mucins as an enzymatically-modulatable barrier to efferocytosis. Together these findings demonstrate a glycocalyx regulatory pathway with implications for therapeutic intervention in the clearance of normal and transformed apoptotic T cells.

T cells die by apoptosis in large numbers during thymic education and following activation and effector function triggered by antigenic challenge[1]. Clearance of apoptotic cells by phagocytes, termed efferocytosis, is a non-immunogenic and non-inflammatory process associated with organismal development, damage resolution, tissue remodeling, repair, and homeostasis[2–4]. Dysregulation of apoptotic cell clearance in general, and T cell clearance in particular, is associated with a range of disorders including autoimmune and inflammatory diseases, viral and bacterial infections, and cancers[4,5]. Phagocytes, primarily macrophages, engage apoptotic target cells using a set of specific receptors and bridging molecules that recognize eat-me

signals amongst which externalized phosphatidylserine (PS)[6] is the best characterized. Eat-me signaling to the phagocyte is counter-balanced by expression of don't eat-me molecules such as CD24, CD31 and CD47 that negatively-regulate phagocytosis through signaling via SHP1 and/or SHP2[7]. Additional cellular barriers to phagocytosis are exemplified by don't-come-close-to-me and don't-eat barriers on the phagocyte, and don't-eat-me structures on the target cell, that negatively-regulate phagocytosis principally via biophysical repulsion[7–11]. T cells and other leukocytes express high densities of transmembrane mucin domain-containing glycoproteins[12,13], which by the nature of their size and negative charge imparted by extensive

[1]The Sir William Dunn School of Pathology, The University of Oxford, Oxford OX13RE, UK. [2]Immunocore Ltd., 92 Park Dr, Milton, Abingdon OX14 4RY, UK. [3]Science for Life Laboratory, Department of Women's and Children's Health, Karolinska Institutet, 17165 Solna, Sweden. [4]SpyBiotech Ltd.; 7600 Quorum, Oxford Business Park North, Oxford OX4 2JZ, UK. [5]Oxford Biomedica plc.; Windrush Court, Transport Way, Oxford OX4 6LT, UK. [6]Max Delbrück Center for Molecular Medicine in the Helmholtz Association; Berlin-Buch, 13125 Berlin, Germany. [7]Experimental and Clinical Research Center (ECRC), Charité Universitätsmedizin Berlin and Max-Delbrück-Center for Molecular Medicine, Lindenberger Weg 80, 13125 Berlin, Germany. [8]These authors contributed equally: Linnea Z. Drexhage, Shengpan Zhang. ✉e-mail: niloofar.karaji@gmail.com; quentin.sattentau@path.ox.ac.uk

sialylation, exhibit broad anti-adhesive properties[14–16] thereby antagonizing phagocytosis[9–11]. We hypothesized that during apoptosis, such a don't eat-me repulsive barrier would need to be efficiently down-modulated to allow optimal recognition of phagocytic eat-me signals and rapid clearance of dying cells. Here we show that during apoptosis, selected mucins within the T cell glycocalyx are cleaved by the ADAM10 sheddase, facilitating T cell efferocytosis by macrophages.

## Results

### Rapid and profound mucin loss from apoptotic T cells

To explore this hypothesis, we screened immortalized and primary human T cells for expression of mucin domain-containing glycoproteins, from hereon called mucins. The immortalized acute T lymphocytic leukemic (T-ALL) cell line CEM expresses a subset of mucins (Supplementary Fig. 1a) including CD43[17], MUC1[18], PSGL-1 (CD162)[19], MUC24 (CD164)[20], and the leukocyte phosphatase CD45 of which all but one isoform contain an N-terminal mucin-like domain[21]. Quantification of cell surface densities using Molecules of Equivalent Soluble Fluorochrome (MESF, Supplementary Fig. 1b) revealed that with the exception of MUC1, all mucins were highly expressed. We induced apoptosis using the protein kinase inhibitor staurosporine, and quantified mucin expression over time by flow cytometry on CEM gated for live, activated caspase-3-positive (apoptotic) and -negative (non-apoptotic) cells within the same starting population (Supplementary Fig. 1c–e). CD43, PSGL-1 and MUC1 surface labeling was

negatively correlated with apoptosis, reducing to <20% of the initial signal by 6 h whereas MUC24 labeling was partially lost (~50% at 6 h) and CD45 remained stable (Fig. 1a). Further quantification was carried out at the 3 h timepoint when ~75% of CEM were apoptotic (Fig. 1a), revealing almost complete loss of CD43, PSGL-1 and MUC1 labeling (all $p < 0.0001$) but minimal MUC24 and no CD45 loss on the apoptotic cells (Fig. 1b). Primary T cells either enriched from human peripheral blood or gated from within monocyte-depleted whole PBMC were activated with either PHA + IL-2 or CD3/CD28 beads + IL-2 to induce MUC1 and MUC24 expression, and staurosporine-treated to induce apoptosis (Supplementary Fig. 1c). Regardless of activation stimulus, both primary CD4+ and CD8+ T cells showed reduced CD43, MUC1, and PSGL1 labeling (all $p < 0.0001$, Fig. 1c and Supplementary Figs. 1f, g, 2a, b). The same pattern of mucin downregulation was also observed in the T-ALL lines Jurkat and HPB-ALL (Supplementary Fig. 2c, d), generalizing this observation in primary and transformed T cells. We also investigated the pre-B-ALL line NALM-6 and the pro-monocytic line U937. Despite a different pattern of basal mucin expression – NALM-6 does not express CD45[22], PSGL1[23] or MUC-1[24] - apoptotic NALM6 showed significantly reduced CD43 compared to its healthy counterpart (Supplementary Fig. 2e), whereas apoptotic U937 showed significantly reduced PSGL-1 and MUC-1 but not CD43, MUC24 or CD45 (Supplementary Fig. 2f). To induce T cell apoptosis in a more physiologically-relevant manner we used the glucocorticoid dexamethasone[25] and observed the same pattern of mucin loss as with

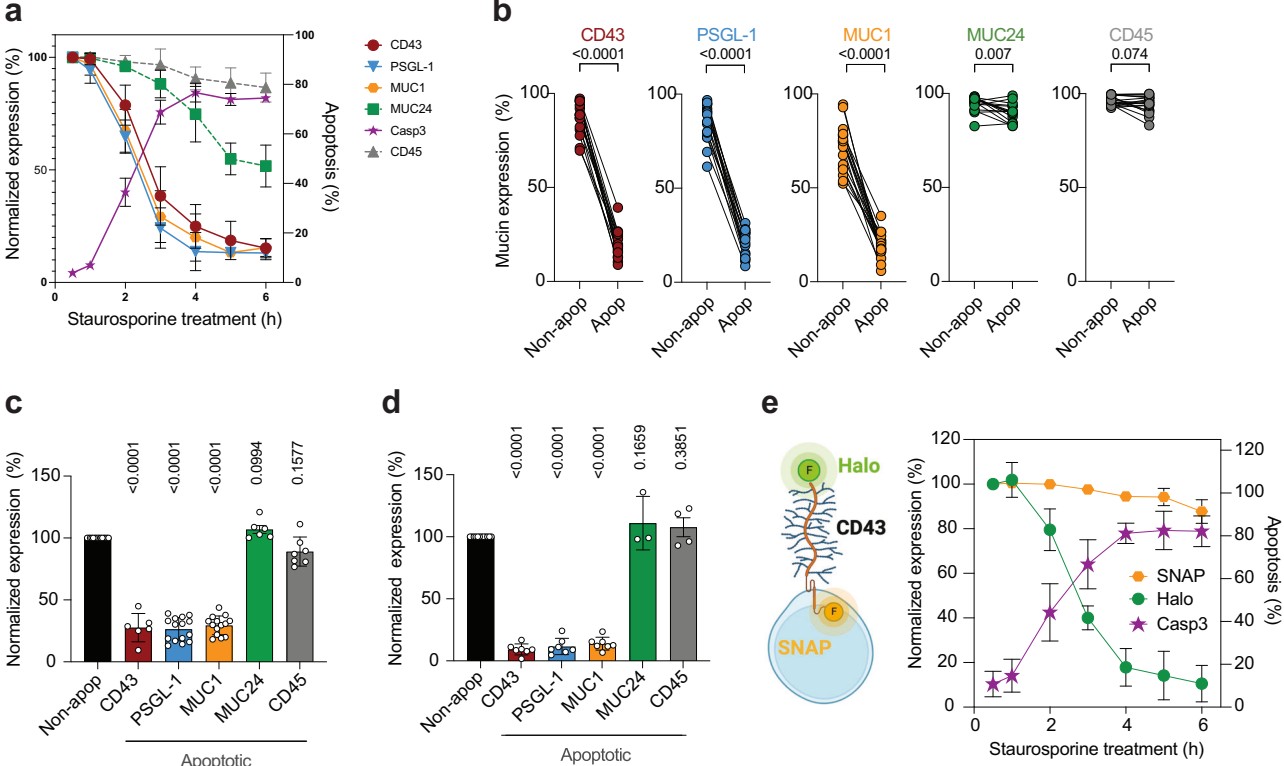

**Fig. 1 | Apoptosis-induced loss of mucin expression from the T cell surface. a**–**e** T cells were treated with staurosporine for 3 h to induce apoptosis, and cell surface mucin expression measured on caspase-3-positive and -negative populations by flow cytometry. **a** Timecourse of mucin expression (left *y*-axis) set to 100% at t = 15 min, right *y*-axis, % cells expressing activated caspase-3 (Casp3, representing apoptosis); *n* = 3–11 independent experiments. **b** Selective loss of mucin expression on CEM cells at t = 3 h analyzed as in (**a**); non-apoptotic population = Non apop; *n* = 8–13 independent experiments. Significance tested using unpaired *t* tests. **c** Mucin expression after apoptosis induction in primary CD4+ T cells, analyzed as in (**a**); *n* = 6–15 independent donors. Significance tested using one-way ANOVA of log₁₀-transformed data with Dunnett's post-hoc test. **d** Mucin expression

after UV-C-induced apoptosis in CEM, data presented as in (**c**); *n* = 3–8 independent experiments. Significance tested using one-way ANOVA of log₁₀-transformed data with Dunnett's post-hoc test. **e** CEM-CD43$_{Halo/SNAP}$ line schematic (created with Biorender.com), and expression of double-labeled CD43-Halo and -SNAP tags over time, expression set to 100% at t = 15 min, *n* = 7 independent experiments. In all graphs, bars or circle fill colors represent: CD43 = burgundy, PSGL-1 = blue, MUC1 = orange MUC24 = green, CD45 = gray, in **a** and **e** Casp3 labeling representing apoptosis = purple and in **c** and **d** data normalized to mucin expression on non-apoptotic (Non-apop) cells set at 100% and shown as a single bar for all analyses. Error bars represent ± 1 SD around the mean. Source data are provided as a Source Data file.

staurosporine in apoptotic CEM, Jurkat and primary CD4[+] T cells (Supplementary Fig. 2g–i). Finally, we excluded off-target pharmacologic effects on mucin labeling by probing cells induced for apoptosis with ultraviolet (UV) radiation. UV-C-induced apoptotic CEM showed the same pattern of downregulated mucin labeling as elicited by staurosporine and dexamethasone, losing CD43, PSGL1 and MUC1 (all $p < 0.0001$), but maintaining CD45 and MUC24 (Fig. 1d). Together these data reveal rapid, substantial and selective loss of mucins from the surface of apoptotic immune cells.

Reduced mucin labeling may result from several non-mutually exclusive mechanisms: (i) alteration of glycan or protein monoclonal antibody (mAb) epitopes; (ii) endocytic internalization; (iii) shedding. Since some anti-mucin mAbs such as anti-CD43 clone DFT-1 are sialic acid-dependent, mAb binding was screened for sensitivity to sialidase treatment. Under conditions where DFT1 labeling was lost, the sialic acid-independent CD43 mAb L10 maintained binding, as did mAbs against MUC1, PSGL1, CD45 and MUC24 (Supplementary Fig. 2j), excluding sialic acid loss as a factor. Modification of CD43 protein epitopes was probed by genetically fusing a *Myc* tag to the extracellular N-terminus of *CD43* (CD43$_{Myc}$), or *Halo* and *SNAP* tags to the *CD43* N- and C-termini respectively (CD43$_{Halo/SNAP}$, Fig. 1e), which were stably expressed in a CD43 knock-out (CD43$_{KO}$) CEM line[26]. MESF analysis revealed similar expression levels of endogenous CD43 and CD43$_{Halo}$, whereas CD43$_{Myc}$ was >2-fold overexpressed (Supplementary Fig. 2k). Staurosporine treatment reduced surface CD43$_{myc}$ and CD43$_{Halo}$ (Supplementary Fig. 2l), excluding mAb epitope modification as the mechanism of reduced labeling. We ruled out CD43 internalization and degradation by inducing apoptosis in CEM$_{Halo/SNAP}$, which triggered progressive loss of CD43$_{Halo}$ but stable CD43$_{SNAP}$ signal, consistent with loss of CD43 ectodomain but retention of the intracellular component (Fig. 1e). We extended our observations to other mucins by treating CEM with concentrations of actin remodeling inhibitors (cytochalasin-D, jasplakinolide) or an endocytosis inhibitor (ikaguramycin) that reduced well-characterized CD71 internalization without influencing caspase-3 activation or cytotoxicity (Supplementary Fig. 3a–c). None of these inhibitors modified apoptosis-induced loss of CD43, PSGL-1 or MUC1 labeling (Supplementary Fig. 3d), excluding mucin endocytosis. Moreover, these results imply that mucin loss is not contingent on actin cytoskeleton remodeling.

## ADAM10 sheds mucins from apoptotic T cells

These findings led us to probe enzymatic cleavage. Since unknown metalloproteases have been implicated in shedding of CD43 on activated leukocytes[27], we tested broad-spectrum metalloprotease inhibitors Marimastat and GM6001, and the ADAM10/ADAM17-specific inhibitor GW280264X (GW). These inhibitors largely preserved CD43, PSGL-1 and MUC1 labeling on apoptotic CEM without influencing MUC24 or CD45 expression or staurosporine-induced caspase-3 activation (Supplementary Fig. 4a, b). Strikingly, the ADAM10-specific inhibitor GI254023X (GI), which as anticipated inhibited cleavage of the well-characterized control substrate CD46[28] (Supplementary Fig. 4c), also reduced loss of CD43, PSGL-1 and MUC1 on apoptotic CEM (all $p < 0.0001$, Fig. 2a) and primary CD4[+] T cells (Supplementary Fig. 4d) without influencing apoptosis (Supplementary Fig. 4b, e). We also confirmed that mucin loss on apoptotic HPBALL, Jurkat, NALM6 and U937 was sensitive to GW and GI treatment (Supplementary Fig. 4f–i), implying that ADAM10 is implicated in mucin cleavage in various immune cell types. To determine whether ADAM10 is necessary and sufficient to mediate mucin loss, we generated *ADAM10* knock-out (A10$_{KO}$) CEM lines using CRISPR-Cas9 (Supplementary Fig. 5a). Apoptotic CEM-A10$_{KO}$ cells shed less CD46 (Supplementary Fig. 5b), and maintained CD43, PSGL-1 and MUC1 expression during apoptosis compared to WT cells (all $p < 0.0001$ Fig. 2b). Similarly, A10$_{KO}$ in CD3/CD28-activated apoptotic primary CD4[+] T cells (Supplementary Fig. 5c, d) largely retained CD43, PSGL-1 and MUC1 (Sup-plementary Fig. 5c, d) largely retained CD43, PSGL-1 and MUC1

compared to apoptotic WT (Supplementary Fig. 5e). Together these data identify ADAM10 as the principal enzyme responsible for mucin loss from apoptotic primary and transformed T cells and a B and monocyte cell line. To investigate whether ADAM10 mediates a baseline level of mucin cleavage in healthy T cells we compared WT CEM or primary T cells with their A10$_{KO}$ counterparts. We observed modestly increased PSGL-1 ($p = 0.026$) and non-significant trends towards increases in CD43, MUC1 and MUC24 expression on A10$_{KO}$ compared to WT CEM (Supplementary Fig. 5f). Primary CD3/CD28-activated CD4[+] T cells showed subtle but significantly increased CD43 ($p = 0.0005$) MUC1 ($p = 0.014$) and MUC24 ($p = 0.025$) expression associated with non-significant trends towards increased PSGL-1 on A10$_{KO}$ cells (Supplementary Fig. 5g). These data reveal limited mucin loss at steady state suggesting tight regulation of ADAM10 mucin shedding activity in healthy T cells.

The ADAM family has functions additional to proteolytic activity such as nuclear signaling[29], which could potentially influence mucin expression independently of sheddase activity. We used fluorescent correlation spectroscopy (FCS) to analyze release of soluble CD43$_{Halo}$ ectodomain in supernatants from CD43$_{Halo/SNAP}$ CEM cells (Fig. 2c). We noted diffusion over time of labeled Halo-ectodomain (Halo$_{A488}$) but not SNAP-intracellular domain (SNAP$_{SiR}$) in apoptotic but not healthy CEM supernatants (Fig. 2d and Supplementary Fig. 6a–c), confirming that these signals were not from apoptotic vesicles containing full-length CD43$_{Halo/SNAP}$. The supernatant Halo$_{A488}$ signal was abolished by ADAM10 inhibitor GI and in A10$_{KO}$ CEM-CD43$_{Halo/SNAP}$ cells, directly linking ADAM10 with CD43 ectodomain shedding (Fig. 2d). Halo$_{488}$ and SNAP$_{SiR}$ were stable and co-diffused, demonstrating intact CD43 in healthy CEM (Supplementary Fig. 6d). Free A488 or Halo$_{A488}$ fragments were excluded by comparing diffusion coefficients of supernatant peaks (snCD43$_{Halo-A488}$) with A488-his-labeled soluble recombinant CD43 ectodomain (srCD43$_{his-A488}$), and free soluble A488 fluorochrome or A488-labeled Halo tag (Halo$_{A488}$). The snCD43$_{HaloA488}$ diffusion coefficient was coordinate with srCD43$_{hisA488}$ but not with free A488 or Halo$_{A488}$ (Fig. 2e), confirming soluble CD43 ectodomain-Halo$_{A488}$ in the supernatant. Final confirmation of CD43 and PSGL-1 ectodomain shedding was obtained by western blotting of immuno-precipitated cell lysates and supernatants from cell surface-biotinylated, apoptotic CEM (Supplementary Fig. 6e). Neutravidin-precipitated lysates and supernatants were analyzed by SDS-PAGE and western blotted with CD43, PSGL-1 and CD45 ectodomain antibodies. MUC1 was not reported because surface expression levels were too low to yield a detectable signal (Supplementary Fig. 1a, b). CEM lysates yielded strong CD43 and PSGL-1 bands that decreased in density after apoptosis induction whereas CD45 expression did not change (Fig. 2f–h), quantified respectively in Fig. 2i–k. By contrast, supernatant band density increased during apoptosis for CD43 and PSGL-1 (Fig. 2f, g), but was not quantified because weak signal-to-noise prevented reproducible outcomes, and no supernatant signal was detected for CD45 under any conditions (Fig. 2h). Apoptosis induction in CEM-A10$_{KO}$ yielded no obvious mucin reduction in lysates or increase in supernatant CD43 or PSGL-1 content by western blotting (Fig. 2f, g, i, j), confirming the central role of ADAM10 in mucin shedding.

## The XKR8 scramblase is required for ADAM10-mediated mucin cleavage

ADAM10 activity is reportedly activated by PS flipping to the outer leaflet, drawing the enzyme active site towards the plasma membrane-proximal substrate via electrostatic interaction between PS head groups and a basic patch on ADAM10[30] (Fig. 3a). However, this mechanism has only been demonstrated in the context of calcium flux-induced PS flipping via TMEM16 family members[31] and we hypothesized that apoptosis might activate ADAM10 via caspase-3-activated scramblase XKR8, implicated in PS flipping within apoptotic cell membranes[32] (Fig. 3a). We therefore knocked-out *XKR8* expression in

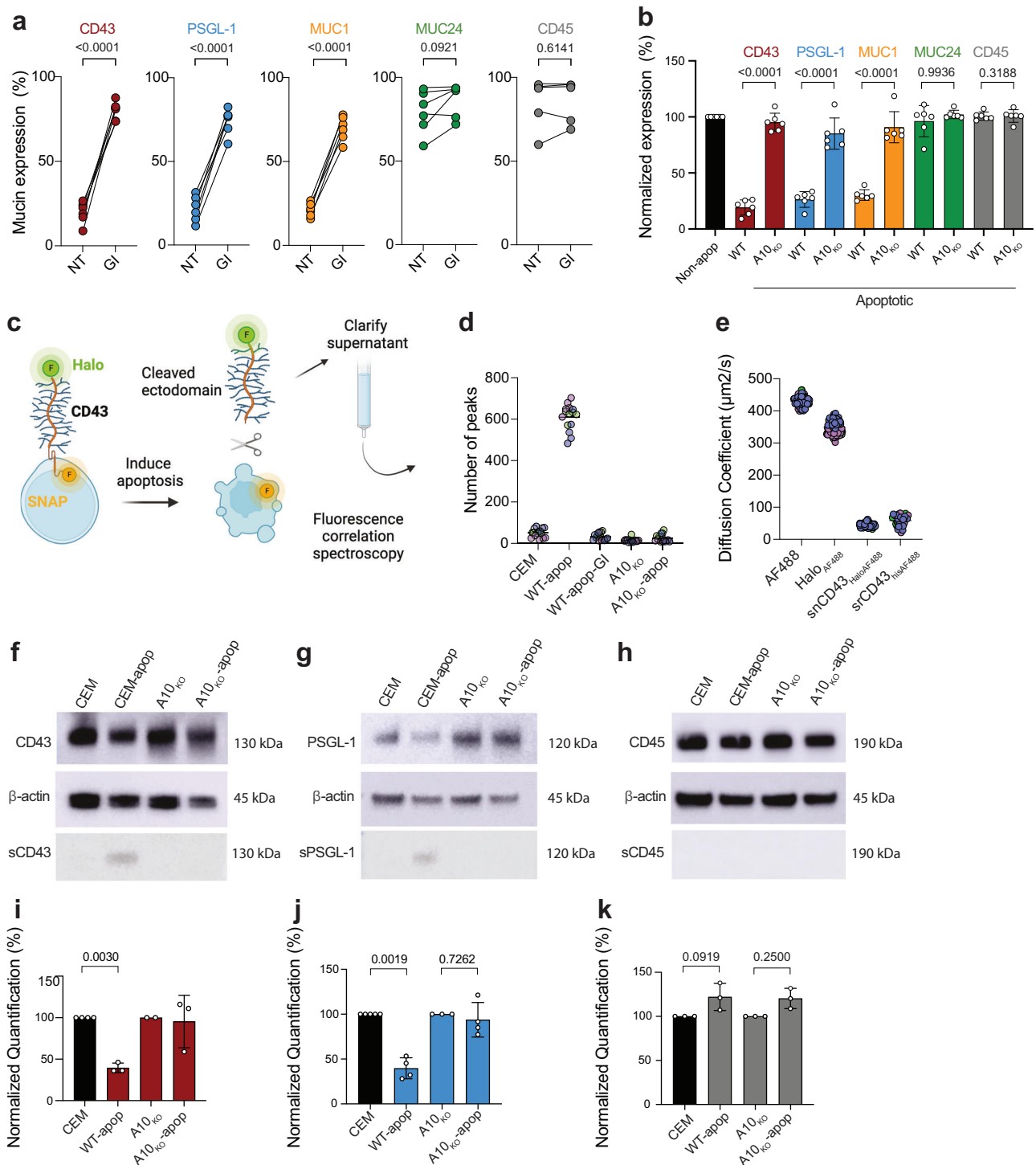

CEM (XKR8$_{KO}$) using CRISPR-Cas9, and screened functionally by inducing apoptosis in single-cell clones and assaying for caspase-3 activation in the absence of PS exposure (Fig. 3b and Supplementary Fig. 7a). XKR8$_{KO}$ was confirmed by sequencing the XKR8 genomic PCR product, which revealed large deletions in 4 independent CEM clones (Supplementary Fig. 7b). Strikingly, apoptosis induction in CEM-XKR8$_{KO}$ cells revealed essentially complete retention of mucins compared to WT cells (Fig. 3c and Supplementary Fig. 7c) supporting the implication of apoptosis-driven XKR8-mediated PS flipping in ADAM10 activation and mucin loss. To exclude non-PS-related effects of XKR8$_{KO}$, we blocked PS headgroups using lactadherin (LA, Fig. 3d), which inhibits ADAM10 activity after TMEM16 flipping of PS[33]. At a non-

toxic concentration which had minimal effects on cell viability and caspase-3 activation (Supplementary Fig. 7d, e) but blocked Annexin-V binding to PS (Fig. 3e), LA significantly reduced loss of CD43, PSGL-1 and MUC1 (Fig. 3f), consistent with XKR8-mediated PS regulation of ADAM10 mucin shedding.

**Mucin loss from apoptotic T cells enhances their efferocytosis**
To directly address the hypothesis that removal of mucins from the dying T cell surface promotes efferocytosis by eliminating a don't-eat-me barrier and potentially thereby better exposing membrane-proximal eat-me signals such as PS, we established a multispectral flow cytometry (ImageStream)-based efferocytosis assay. Monocyte-

**Fig. 2 | ADAM10 selectively sheds mucins. a** Pre-treatment of CEM with ADAM10 inhibitor GI prior to staurosporine-induced apoptosis and mucin analysis; non-drug-treated cells = NT $n = 6$–8 independent experiments. Significance tested using unpaired $t$ tests. **b** ADAM10-KO (A10$_{KO}$) CEM line analyzed for mucin expression, non-apoptotic cells (Non-apop) normalized to 100% and represented as a single bar for all groups; $n = 6$ independent experiments. Significance tested using unpaired $t$ tests of log$_{10}$-transformed data. **c** Schematic of approach to detect shed CD43 ectodomain in apoptotic CEM$_{Halo/SNAP}$ supernatants using fluorescence correlation spectroscopy (created with Biorender.com). **d** Differential diffusion of soluble CD43 ectodomain in supernatants from apoptotic WT CEM (WT-apop) but not apoptotic WT CEM treated with GI (WT-apop-GI) or apoptotic ADAM10-KO (A10$_{KO}$-apop). **e** Controls for CD43 ectodomain shedding. Diffusion of AF488 = free fluorochrome; Halo$_{AF488}$ = fluorochrome-labeled soluble Halo tag; snCD43$_{HaloAF488}$ = free CD43 ectodomain released from CEM$_{Halo/SNAP}$ into supernatant; srCD43$_{hisAF488}$ = soluble recombinant A488-labeled his-tagged CD43 ectodomain. In **d** and **e** circles represent 10 replicate measurements within a single experiment, different fill colors represent individual experimental repeats ($n = 3$). **f**–**h** western blot of CEM cell lysates (top), β-actin loading control (middle) and supernatants (lower) from healthy and apoptotic WT or A10$_{KO}$ CEM. **f** CD43, **g** PSGL-1, **h** CD45. **i**–**k** Densitometric quantification of western-blotted mucin bands from lysates of non-apoptotic and apoptotic WT and A10$_{KO}$ CEM, band density normalized to β-actin loading control and non-apoptotic cell lysate signal intensity set to 100%; $n = 2$–5 independent experiments. Significance tested using paired $t$ tests of log$_{10}$-transformed data. **i** CD43; **j** PSGL-1; **k** CD45. In **a**, **b**, and **i**–**k**, bar or circle fill colors represent: CD43 = burgundy, PSGL-1 = blue, MUC1 = orange MUC24 = green, CD45 = gray, and in **b** and **i**–**k**, data normalized to mucin expression on non-apoptotic (Non-apop) cells set at 100% and shown as a single black bar. Error bars represent ±1 SD around the mean. Source data are provided as a Source Data file.

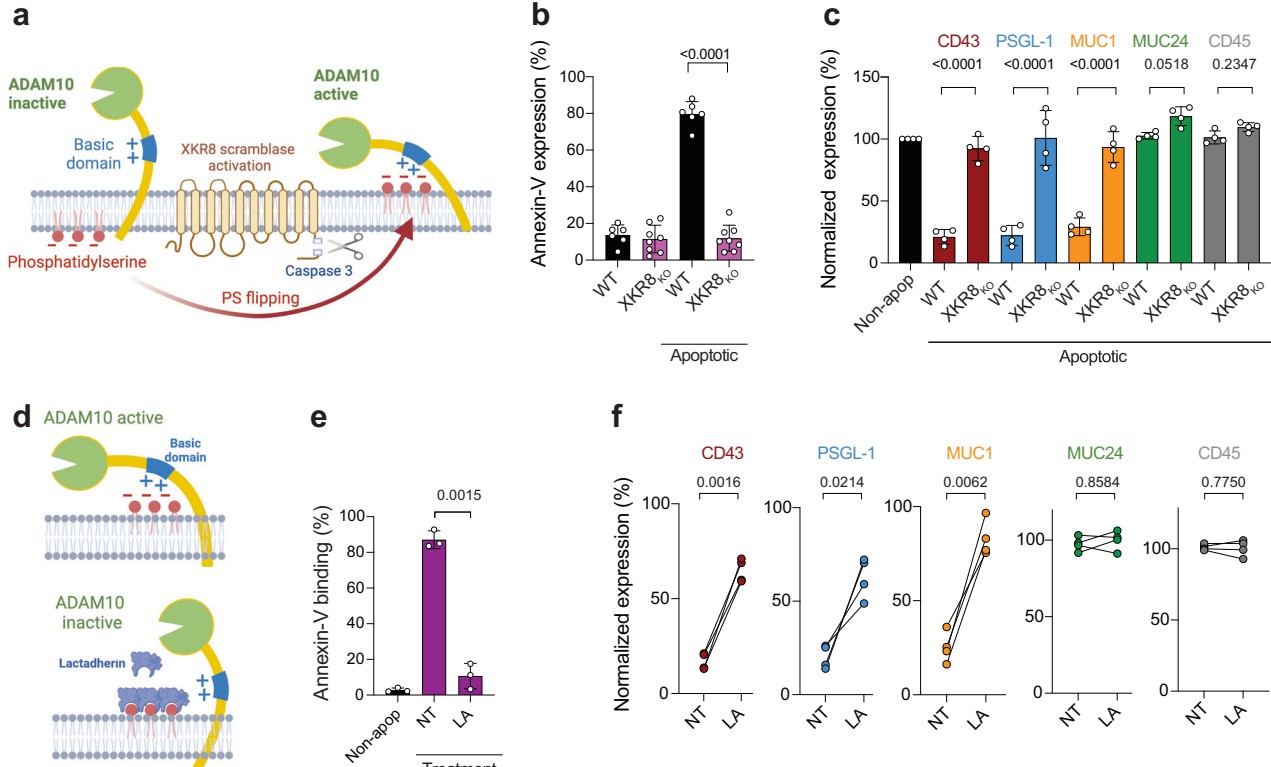

**Fig. 3 | XKR8 and PS-mediated activation of ADAM10 is required for loss of mucin expression. a** Schematic of putative mechanism of ADAM10 activation for mucin cleavage (created with Biorender.com): scramblase XKR8 is activated by caspase-3 and flips PS from the inner to the outer plasma membrane leaflet during apoptosis. PS on the outer plasma membrane leaflet triggers ADAM10 sheddase activity. **b** XRK8 knocked-out in CEM using CRISPR-Cas9 (XKR8$_{KO}$), and WT and XKR8$_{KO}$ clones exposed to staurosporine, apoptotic cells gated on caspase-3 activation and PS expression detected by annexin-V labeling, $n = 6$–8 independent experiments. Bar fill colors represent WT CEM = black, XKR8$_{KO}$ = purple. Significance tested using an unpaired $t$ test. **c** Staurosporine-treated XKR8$_{KO}$ CEM gated on activated caspase-3 to define apoptotic and non-apoptotic cells were labeled for mucin expression, data normalized to mucin expression on non-apoptotic (black bar) cells set to 100%, $n = 4$ independent experiments, bar fill colors represent CD43 = burgundy, PSGL-1 = blue, MUC1 = orange, MUC24 = green, CD45 = gray. Significance tested using paired $t$ tests. **d** Schematic model for Lactadherin (LA) inhibition of PS-mediated ADAM10 activation (created with Biorender.com). **e** LA inhibition of annexin-V binding to PS on activated caspase-3$^+$ apoptotic CEM cells, $n = 3$ independent experiments. Bar fill colors represent non-apoptotic (Non-apop) CEM = black, No treatment (NT) and lactadherin (LA)-treated = purple. Significance tested using a paired $t$ test. **f** LA inhibition of mucin cleavage on apoptotic CEM cells, $n = 4$ independent experiments. Filled circles represent CD43 = burgundy, PSGL-1 = blue, MUC1 = orange MUC24 = green, CD45 = gray. Significance tested using paired $t$ tests of log$_{10}$-transformed data. Error bars represent ±1 SD around the mean. Source data are provided as a Source Data file.

derived macrophages (MDM) isolated and differentiated from human peripheral blood mononuclear cells (PBMC) were labeled, cocultured with apoptotic CEM or primary T cells, lifted and washed in PBS-EDTA to remove unengulfed T cells, fixed and permeabilized, stained for cleaved caspase-3 and analyzed for T cell uptake (Supplementary Fig. 8a). In-focus events, double-labeled for MDM and T cells, were gated using the delta-centroid function that measures T cell proximity to the MDM center, corresponding to T cell attachment or uptake (Fig. 4a and Supplementary Fig. 8b). We confirmed that this assay did indeed detect phagocytosed T cells by inhibiting actin cytoskeleton remodeling and phagocytosis with jasplakinolide, cytochalasin D or latrunculin[34,35]. These agents strongly reduced both apoptotic CEM (Supplementary Fig. 8c) and control bead uptake (Supplementary Fig. 8d) in the absence of obvious toxicity (Supplementary Fig. 8e, f).

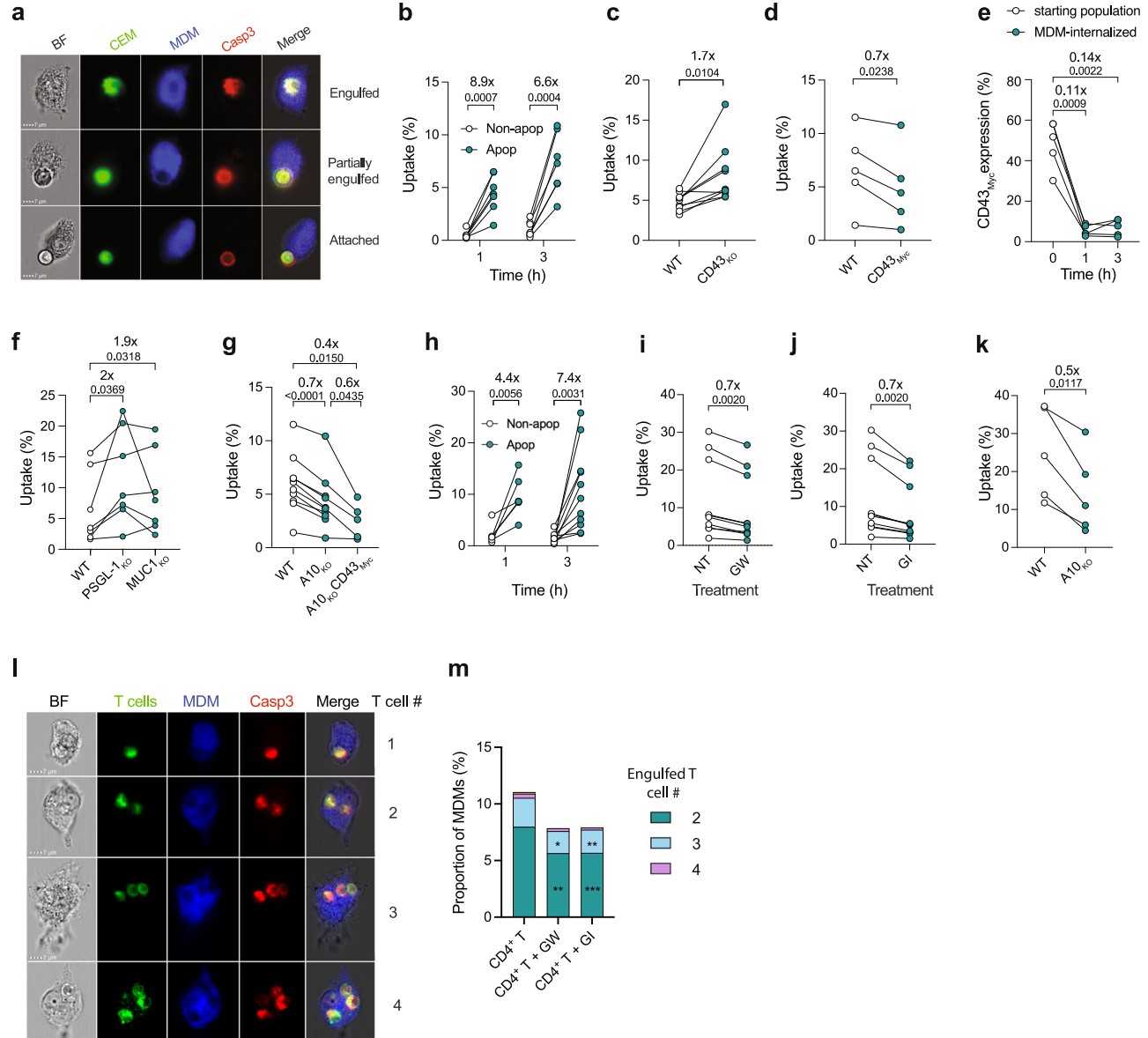

**Fig. 4 | ImageStream-based analysis of apoptotic T cell uptake by MDM.**
**a** Representative images of activated caspase-3⁺ (Casp3) apoptotic CEM within or adjacent to MDM following 1 h coculture, scale bar = 7 μm. **b** Non-apoptotic (Non-apop, open circles) or apoptotic (Apop, green filled circles) WT CEM analyzed for MDM uptake after 1 or 3 h coculture by ImageStream, $n = 6$ independent experiments. Significance tested using paired $t$ tests. **c** Apoptotic WT (open circles), CD43$_{KO}$ (green filled circles), $n = 7$ independent experiments. Significance tested using paired $t$ tests. **d** Apoptotic WT (open circles), CD43$_{KO}$ overexpressing CD43-Myc (CD43$_{Myc}$, green filled circles), $n = 5$ independent experiments analyzed for uptake after 1 h coculture with MDM. Significance tested using paired $t$ tests.
**e** Comparison of the starting CEM culture (open circles), with MDM-engulfed CEM (green filled circles), calculated by the ratio of % Myc⁺ CEM in the starting culture with % Myc⁺ CEM within MDM, $n = 4$ independent experiments. Significance tested using paired $t$ tests. **f** Relative MDM uptake of apoptotic WT CEM (open circles) or CEM knocked out for PSGL-1 or MUC-1 (PSGL-1$_{KO}$ and MUC1$_{KO}$ respectively, green filled circles); $n = 7$ independent experiments. Significance tested using paired $t$ tests. **g** Relative uptake of apoptotic WT CEM (open circles), CEM knocked out for ADAM10 or ADAM10 knock-out overexpressing CD43$_{Myc}$ (A10$_{KO}$ and

A10$_{KO}$CD43$_{Myc}$ respectively, green filled circles); $n = 9$ independent experiments. Significance tested using paired $t$ tests. **h** Non-apoptotic (Non-apop, open circles) or apoptotic (Apop, green filled circles) primary CD4⁺ T cells analyzed for MDM uptake after 1 or 3 h coculture as in (**b**), $n = 5$–11 independent donors. Significance tested using paired $t$ tests on log$_{10}$-transformed data (1 h) or non-transformed data (3 h). **i** MDM uptake of untreated (NT, open circles), GW-treated or **j** GI-treated primary T cells (green filled circles) after 3 h coculture, $n = 10$ independent donors. Significance tested using Wilcoxon signed-rank test. **k** MDM uptake of apoptotic WT (open circles) or A10$_{KO}$ (green filled circles) primary T cells, $n = 4$ independent donors. Significance tested using paired $t$ test. **l** Representative images of activated caspase-3⁺ (Casp3) apoptotic primary T cells engulfed by MDM at t = 3 h.
**m** Percentages of MDM with >1 engulfed primary T cell, untreated or treated with GW or GI, $n = 1000$ independent images analysed, MDM containing one T cell excluded to prioritize presentation of multiple uptake events. Numbers above bars represent fold-difference in T cell uptake by MDM, 2 cells = green, 3 cells = light blue, 4 cells = purple. Error bars represent ±1 SD around the mean. Significance tested using paired $t$ tests, CD4⁺ T + GW, *$p = 0.035$, **$p = 0.0018$; CD4⁺ T + GI, **$p = 0.003$, ***$p = 0.0007$. Source data are provided as a Source Data file.

We calculated % MDM containing one or more apoptotic T cells, and since independent T cell-MDM cocultures had different starting proportions of apoptotic: non-apoptotic T cells, we normalized between experiments by expressing uptake data as [% MDM-engulfed apoptotic T cells / % apoptotic T cells prior to coculture with MDM] x 100. After 1 and 3 h coculture, CEM showed 8.9- and 6.6-fold differential MDM uptake respectively between non-apoptotic and apoptotic cells, demonstrating selective MDM capture of apoptotic cells (Fig. 4b). We analyzed the early contribution of mucins to efferocytic recognition of apoptotic T cells by comparing WT CEM with CD43$_{KO}$, with the anticipation that absent expression of mucin at the outset of CEM-MDM coculture might accelerate MDM engagement. Using the Imagestream delta-centroid function to analyze T cell attachment to the MDM surface (Supplementary Fig. 8b), we found that apoptotic CEM-CD43$_{KO}$ showed increased attachment of 1.9- and 2.5-fold after 1 and 3 h coculture, respectively, compared to apoptotic CEM$_{WT}$, (Supplementary Fig. 9a). This translated into MDM uptake of apoptotic CEM cells, since engulfment of CEM-CD43$_{KO}$ was 1.7-fold (Figs. 4c) and 1.2-fold (Supplementary Fig. 9b) more at 1 and 3 h respectively than CEM$_{WT}$. By contrast, over-expression of CD43$_{Myc}$ in CD43$_{KO}$ cells reduced MDM uptake to 0.7-fold at 1 (Fig. 4d) and 3 h (Supplementary Fig. 9c). Comparing % CEM overexpressing CD43$_{Myc}$ pre-MDM coculture with those engulfed by MDM after 1 and 3 h revealed 9- and 7-fold increases of CD43$_{Myc}$-depleted cells within MDM, respectively (Fig. 4e), presumably reflecting reduced uptake of CD43$_{Myc}$-high expressing cells in favor of CD43$_{Myc}$-depleted cells. To further investigate the role of mucins during early efferocytosis recognition events in apoptotic cells, we used CRISPR-Cas9 to prepare CEM clones lacking expression of *SELPLG* encoding PSGL-1 (PSGL-1$_{KO}$, Supplementary Fig. 9d) and *MUC1* (MUC1$_{KO}$, Supplementary Fig. 9e). PSGL-1$_{KO}$ and MUC-1$_{KO}$ CEM were taken up 2-fold and 1.9-fold respectively more than WT (Fig. 4f), confirming the relevance of these mucins to regulating macrophage efferocytosis of apoptotic cells.

### ADAM10 regulates efferocytic uptake of apoptotic T cells

Since ADAM10 sheds mucins, we tested CEM-A10$_{KO}$ for efferocytic uptake. Compared to CEM$_{WT}$, CEM-A10$_{KO}$ reduced MDM uptake, which was further reduced by overexpression of CD43$_{Myc}$ (Fig. 4g), with a similar pattern after 3 h MDM coculture (Supplementary Fig. 9f). Transformed cells differentially express and regulate molecules implicated in efferocytosis compared to normal cells[4,36]. We, therefore, analyzed the role of ADAM10-mediated mucin modulation of efferocytosis in primary CD4$^+$ T cells. Apoptotic primary T cell uptake by MDM was 4.4- and 7.4-times more likely than their non-apoptotic counterparts at 1 and 3 h post-coculture (Fig. 4h), but was significantly less after pre-treatment with inhibitors GW and GI (Fig. 4i, j, respectively). Consistent with pharmacological inhibition of ADAM10, apoptotic primary T cells knocked out for ADAM10 (Supplementary Fig. 5c, d) were 2-fold less taken up than their WT counterparts (Fig. 4k), confirming the influence of ADAM10 on efferocytosis in both immortalized and primary T cells. Macrophages can phagocytose multiple apoptotic cells[37], and a fraction of MDM engulfed >1 primary T cells (Fig. 4l), with GW and GI pre-treatment significantly reducing the fraction of MDM containing multiple cells (Fig. 4m). Despite their relatively large diameter, multiple CEM were also engulfed by a small fraction of MDM, and A10$_{KO}$ and overexpression of CD43$_{Myc}$ reduced that fraction (Supplementary Fig. 10).

### Discussion

Our results are consistent with a model in which selected mucins within the T cell glycocalyx act as a don't get close to me and/or don't-eat-me barrier that broadly inhibits phagocytosis and efferocytosis, and which is rapidly and profoundly downregulated during apoptosis induction by ADAM10 via PS flipping to the plasma membrane outer leaflet. Loss of this extended barrier would not only remove the

biophysical constraints inhibiting close cell-cell contact[7,10,11], but would enhance PS exposure and potentially other membrane-proximal eat-me signals for receptor engagement (Supplementary Fig. 11). Our study links the following events: (i) selected mucin shedding triggered by apoptosis; (ii) regulated loss of mucins enhancing efferocytosis; (iii) the pathway implicating caspase-3-activated XKR8 to trigger ADAM10 sheddase activity. Whilst this analysis is in vitro, it is interesting to note that transmembrane mucins such as CD43 are shed from cell membranes under inflammatory conditions in vivo, and their extracellular domains can be detected in biological fluids[12,13,38].

Whilst the current study is primarily limited to T cells, other cell types including those of immune, epithelial and endothelial origin express high levels of mucins within their glycocalyx which may be subject to enzymatic regulation. In this respect, we show that B cell and myeloid cell lines also selectively downmodulate mucins in response to apoptosis induction, although in a pattern distinct from that observed in T cells. A likely explanation for this cell type-selective ADAM10 activity is the differential expression of tetraspanins that precisely modulate ADAM10 substrate specificity[39,40]. Here, we only investigate apoptosis which is the dominant paradigm for PS externalization leading to efferocytosis. However, other types of regulated cell death such as necroptosis, pyroptosis and ferroptosis are also reported to result in PS externalization[41] with the potential to activate ADAM10, shed mucins and promote phagocytic clearance. Furthermore, PS flipping also occurs by other mechanisms in the absence of cell death, as in the case of calcium-mediated activation of TMEM16F[41,42]. In this respect, it is interesting to speculate that calcium flux elicited during T cell activation[43] may trigger ADAM10 activity resulting in loss of mucins, potentially facilitating T cell interactions with antigen presenting cells.

Understanding the regulation of mucin expression in apoptotic cells has obvious implications for disease processes. Dysregulation of mucin downregulation will have important downstream consequences for removal of dying cells, with the potential for their pathological accumulation[2,4]. Conversely, inappropriate activation of ADAM10 may promote unwanted interaction of phagocytes with healthy cells leading to uncontrolled engagement and engulfment. Indeed, dysregulation of apoptotic T cell clearance is associated with a range of immune disorders including autoimmune and inflammatory diseases and viral and bacterial infections[5]. Interestingly, a number of immune pathologies including systemic lupus erythematosus, atherosclerosis and rheumatoid arthritis associated with defects in efferocytosis[4], are independently associated with ADAM10 function and dysfunction[44,45], suggesting possible causal links. Since metalloproteinases including ADAM family members are similarly expressed in multiple tissues, mucin modulation through shedding may be a phenomenon more generally associated with clearance of apoptotic cells. Thus, our findings introducing ADAM10 as a central regulator of mucin expression on T cells during apoptosis may suggest potential therapeutic avenues for modulating efferocytic clearance of immune cells.

### Methods

#### Inclusion and ethics statement

Leukocyte cones from healthy donors were provided by the National Health Service UK (NHS) Blood and Transplant services (Oxford, UK) under contract 17/WM/0333. Informed consent was obtained from each donor before blood collection.

#### Cell lines

Acute T cell lymphoid leukemia (T-ALL) lines CEM[46] and the CD43-KO counterpart (CD43$_{KO}$)[26] were kindly provided by the Ardman lab, and Jurkat E6.1[47] and HPBALL[48], and the pre-B-ALL line NALM-6[49] and pro-monocytic line U937[50] lines were obtained from the Sir William Dunn School of Pathology cell bank. All leukocyte lines were grown in complete RPMI (ThermoFisher 11875093) containing 10% FCS (Sigma-

Aldrich F9665), 1% penicillin/streptomycin (ThermoFisher 15140122) at 37 °C/5% $CO_2$. HEK 293 T cells were grown in complete DMEM (ThermoFisher 12491023) 10% FCS, 1% penicillin/streptomycin, 1% HEPES (ThermoFisher 15630056) at 37 °C/5% $CO_2$.

### Isolation and activation of primary T cells from PBMCs
Peripheral blood mononuclear cells (PBMC) were obtained by gradient centrifugation on Ficoll-HistoPaque 1077 (Sigma-Aldrich 10771). After red cell lysis in ACK lysis buffer (ThermoFisher A1049201) the PBMC fraction was incubated in 10 $cm^2$ tissue culture petri-dishes (Corning CLS430167) at $150 \times 10^6$ cells/dish for 2 h at 37 °C/5% $CO_2$ in complete RPMI-1640 to separate adherent cells, and plates washed once with warm PBS (ThermoFisher 10010001) to remove non-adherent lymphocyte-containing fractions. $CD4^+$ T cells were isolated from monocyte-depleted PBMCs using EasySep Human $CD4^+$ Selection Kit II (StemCell Technologies 19662) according to manufacturer's instructions. Monocyte-depleted PBMCs or isolated $CD4^+$ T cells were stimulated immediately after their respective isolation with PHA (1 μg/mL, Sigma-Aldrich 11082132001) or Dynabeads Human T-Activator CD3/CD28 (ThermoFisher 11131D), cultured in recombinant human IL-2 (10 U/mL, NIBSC 86/500) in complete RPMI and used between days 5–7.

### Monocyte-derived macrophage preparation
Plastic adherent monocytes derived from PBMC (above) were cultured in complete RPMI supplemented with recombinant human M-CSF (10 ng/mL, Gibco AF-300-25) for 7–10 days to differentiate monocyte-derived macrophages (MDM). After 7–10 days differentiation, MDM were detached with trypsin-EDTA (Gibco 11580626) and gentle scraping, and re-plated at $5 \times 10^6$ cells per 6 cm petri dish (Corning CLS430165) at least 24 h prior to coculture with isolated $CD4^+$ T cells for the efferocytosis assay.

### Apoptosis induction in cell lines and primary T cells
Cell lines, purified primary $CD4^+$ T cells or monocyte-depleted PBMCs were counted and resuspended at a concentration of $2–3 \times 10^6$ cells/mL in complete RPMI. Staurosporine (Cambridge Bioscience S-7600) pre-titrated to give ~50–80% apoptosis at 3 h post-treatment, which for most experiments was 10 μM, was added for the indicated time (from 30 min to 6 h) at 37 °C. For dexamethasone-induced apoptosis induction, dexamethasone (10 μM, ThermoFisher A13449) or carrier only (DMSO Sigma-Aldrich D4540) was added for 48 h. At the appropriate time, cells were washed with PBS and collected for further manipulation. UV-C radiation[51] was used to induce apoptosis in CEM cells by radiation at 50 $J/m^2/s$ for 2 min. Cells were incubated for 24 h and subsequently prepared for flow cytometric analyses.

### Flow cytometry analysis of healthy and apoptotic T cells
For analysis of cell phenotype during apoptosis, T cell lines, monocyte-depleted PBMCs or purified primary $CD4^+$ T cells were centrifuged for 5 min at $400 \times g$. Pellets were resuspended in 100 μL cold annexin V binding buffer (BD Pharmingen 556454) for 20 min at 4 °C in the dark with annexin V-FITC or -pacific blue (Biolegend 640906 or 640918, respectively) used at 1:100, near-IR fixable viability dye (Invitrogen L10119) at 1:1000 and fluorophore-conjugated primary antibodies including: anti-human CD43 sialic acid-independent clone L10-APC[52] (Invitrogen 10746863) and sialic acid-dependent clone DFT-1[53]-APC (Becton-Coulter B49195), -PE (Becton-Coulter A32560) or -AF647 (Santa-Cruz sc6256) all used at 2 μg/mL; anti-human CD45-PE (MEM-28, Abcam ab134202) used at 1:100; anti-human MUC1-PE clone 16 A (Biolegend 355604) used at 2 μg/mL; anti-human MUC24-PE clone 67D2 (Biolegend 324808) used at 0.5 μg/mL; anti-human PSGL-1-PE clone TC2 (Invitrogen MA1-10117) used at 1:300; anti-human ADAM10-BV421 clone 11G2 (BD Biosciences 742787) used at 1:100. In the case of monocyte-depleted PBMCs, anti-human CD4 clone

RAPA-T4 on Percp-Cy5.5 (Biolegend, 300530) used at 1 μg/mL, and anti-human CD8 clone SK1 on AF700 (Biolegend 344724). All labeling was done in conjunction with the corresponding concentration-matched isotype control antibody. After labeling, cells were washed with cold annexin V binding buffer where appropriate, centrifuged for 2 min at $400 \times g$ at 4 °C, and fixed with 4% paraformaldehyde (Sigma-Aldrich 158127) for 10 min at RT. Cells were washed with PBS and permeabilized with perm buffer (Biolegend 421002), and labeled with anti-human active caspase-3 antibody clone Asp175 (Cell Signaling Technologies 9661) used at 1:400. After 30 min incubation at 4 °C, cells were washed with FACS wash buffer (PBS, 2% FCS) and incubated with anti-Rabbit IgG (H + L) Alexa Fluor 647, (Invitrogen, A21245) or Alexa Fluor 546 (Invitrogen, A11010), both used at 4 μg/mL (1:500) where required 30 min at 4 °C. After washing, cells were analyzed by flow cytometry using a Cytoflex LX flow cytometer (Beckman Coulter) and data processed using the FlowJo-V10 software (FlowJo, LLC). Isotype controls and Fluorescence Minus One (FMO) controls were performed for all colors to gate on positive and negative populations. Gating on the relevant cell population was set according to Forward Scatter (FSC) and Side Scatter (SSC) before doublet and near-IR fixable viability dye (Thermo Fisher) to allow exclusion of dead cells with dye-permeable membranes. Where monocyte-depleted PBMC samples were used, gates were set to differentiate $CD4^+$ and $CD8^+$ T cell subsets. Subsequent gating for all samples was carried out on annexin V and/or activated caspase-3 labeling to differentiate apoptotic from non-apoptotic cells within the same total cell population.

### Molecules of Equivalent Soluble Fluorochrome (MESF) Quantification
$2–3 \times 10^6$ healthy cells were plated and stained with 1 μg PE-labeled antibody specific for CD43, PSGL-1, MUC1, MUC24 or CD45 for 30 min at 4 °C in the dark. Cells were centrifuged for 2 min at $400 \times g$ at 4 °C and resuspended in FACS buffer. PE-beads (Quantum R-PE MESF, Bangs Laboratories 827) were prepared according to the manufacturer's protocol and run immediately together with the samples on a Cytoflex LX. Data were analyzed using the manufacturer's software QuickCal v 3.0.

### Metalloprotease inhibitors
Broad spectrum metalloprotease-specific inhibitors (GM6001, R&D Systems 2983; Marimastat, R&D Systems 2631) or ADAM10 inhibitor GI254023X, here called GI[54] (Aobious AOB3611) or ADAM10/17 inhibitor GW280264X, here called GW[54] (Aobious AOB3632) were each added at the pre-determined optimum concentration of 10 μM for 40 min prior to the addition of staurosporine or dexamethasone, and maintained during the experiment. Where UV-C light was used to induce apoptosis in CEM, cells were incubated with GI and GW prior to irradiation as described above and UV-sensitive GW was added again at 10 μM after irradiation.

### CD71 internalization assay
CEM were counted and resuspended at a concentration of $2–3 \times 10^6$ cells/mL in ice-cold FACS wash buffer (PBS, 10% FCS, 2 mM EDTA) containing 1.25 μL anti-human CD71 clone OKT9 on FITC (Invitrogen 14-0719-82) used at 0.6 μg/mL for 1 h at 4 °C then moved to 37 °C and incubated for 1 h with or without 100 μM cytochalasin D (Sigma-Aldrich C8273), 100 μM jasplakinolide (Abcam ab141409) or 1 μM ikarugamycin[55] (Abcam ab143408). Cells were acid-washed (0.1 M glycine and 150 mM NaCl at pH 3) for 1 min, washed twice in cold FACS buffer, and labeled with near-IR fixable viability dye used at 1:1000 (Invitrogen L10119) before fixation and flow cytometric analysis. Gating on the relevant cell population was set according to Forward Scatter (FSC) and Side Scatter (SSC) before doublet and dead cell exclusion. Internalization was represented by the geometric mean

fluorescence intensity (GMFI) of CD71 normalized to non-internalized sample.

## Lactadherin PS binding assay

WT CEM were resuspended at a concentration of $2-3 \times 10^6$ cells/mL in complete RPMI. Lactadherin (LA, Prolytix BLAC-1200) was added at a pre-determined optimum concentration of $2\,\mu M$ that inhibits binding of annexin V to PS but is not toxic for 40 min prior to the addition of staurosporine, and maintained during the experiment. Cells were then harvested, stained for mucins or annexin V, fixed, permeabilized, labeled for activated caspase-3, and analyzed by flow cytometry.

## ADAM10, PSGL-1, MUC1 and XKR8 knock-out in CEM

WT CEM were resuspended at $2 \times 10^7$/mL in buffer R (ThermoFisher BR5). Guide RNA (gRNA) sequences for *SELPLG* encoding PSGL-1 (https://www.ensembl.org/Homo_sapiens/Gene/Summary?db=core;g = ENSG00000110876): ACCACGGAGCUGGCC, GACCACUC AACCAGU, *CACUCAACCUGUGCC. Sequences of gRNA for MUC1* (https://www.ensembl.org/Homo_sapiens/Gene/Summary?db=core;g=ENSG00000185499): GCUACGAUCGGUACU, GCAACAGUU-GUUACG, GGUCAUGCAAGCUCU, CAUGCAAGCUCUACC. Sequence of gRNA for the *ADAM10* gene (NP_001101.1, https://www.ncbi.nlm.nih.gov/protein/NP_001101.1) for the line of KO ADAM10 KO cells analysed here out of three successful KO lines made: AAAATGTGCCACCAC-GAGTC. Sequences of gRNA for the *XKR8 gene* https://www.ensembl.org/Homo_sapiens/Gene/Summary?db=core;g=ENSG00000158156): CCTACGCCGACTTCCTCGCCCTG, CCTTGCGCACCTGCCTCCCCTCC, CCTCCGTGATCTACTTCCTGTGG. Each gRNA ($4.4\,\mu L$ of 100 pmol/$\mu L$, IDT) was mixed with $4.4\,\mu L$ tracrRNA (Alt-R CRISPR-Cas9 tracrRNA, 20 nmol, IDT) and $1.2\,\mu L$ Duplex buffer and boiled at $95\,°C$ for 5 min. Ribonucleoproteins (RNPs) were prepared by incubating $0.6\,\mu L$ RNA mix, $0.6\,\mu L$ Cas9 (Alt-R S.p. HiFi Cas9 Nuclease V3, $100\,\mu g$ (IDT 1081058) at RT for 20 min. Post-incubation, $10\,\mu L$ of cells, $2\,\mu L$ transfection enhancer (Alt-R Cas9 Electroporation Enhancer, 10 nmol, IDT) and $1\,\mu L$ RNPs were mixed and immediately taken up with a Neon Transfection System pipette tip (ThermoFisher MPK1025K). The pipette was inserted into the Neon Transfection Machine (Neon1 ThermoFisher), containing 3 mL buffer E (ThermoFisher MPK1025K). Transfection conditions were adapted to CEM cells at 1230 V, 40 ms pulse width, 1 pulse. After transfection, cells were resuspended in R10 without antibiotics. After 48 h, penicillin and streptomycin were added and knock-out efficiencies assessed by flow cytometry. Cell sorting was carried out on the single, live and marker negative population of each clone using the Aria III cell sorter (Becton-Dickinson). Single cells sorts were done for PSGL-1, MUC1 and XKR8 knock-outs. Subsequent ADAM10$_{KO}$, PSGL-1$_{KO}$ and MUC1$_{KO}$ cell lines were confirmed by western blot with anti-ADAM10 polyclonal (Merck AB19026) at 1/500 followed by goat anti-rabbit HRP (BioRad 1706505) used at 1/1000, anti-PSGL-1 clone 108 (Thermo Fisher MA5-29555) at 1/500 followed by goat anti-rabbit HRP (BioRad 1706505) used at 1/1000, anti-MUC1 clone MH1-CT2 (ThermoFisher MA5-11202) at 1/500 followed by anti-Armenian hamster-HRP (ThermoFisher PA1-32045) at 1/1000. Because we were unable to validate antibodies specific for XKR8, selection of XKR8 KO clones was done by single cell sorting of single, live and annexin V negative cells, followed by expansion and analysis of clones for lack of PS flipping to the outer membrane of staurosporine-induced apoptotic cells that were positive for intracellular activated caspase-3 labeling. Cells positive for activated caspase-3 but negative for PS flipping were subsequently grown up for further analysis. In the absence of a validated XKR8-specific commercial antibody we could not verify protein content by western blotting. We therefore characterized the KO by sequencing the *XKR8* gene PCR product from individual clones using primers specific for amplification of Exon 3. Genomic DNA was purified from single cell clones using the Monarch Genomic DNA Purification Kit (New England Biolabs T3010S) and Exon

3 amplified using Q5 High-Fidelity DNA polymerase (M0491S New England Biolabs). The following primers were used: 5′-3′ ACCTGTGACCGCTGGGGAGT, 3′-5′ TGGGTCTCTACAAGTGACA-GATGTGTTG. The PCR products were run on a 1% agarose gel with SYBR-Safe DNA gel stain (Invitrogen S33102). The bands were subsequently gel-purified using the QIAquick gel extraction kit (Qiagen 28704) and sequenced by Source BioScience. Sequence analysis and alignment was done using SnapGene and Jalview software.

## ADAM10 knock-out in primary CD4$^+$ T cells, cell expansion and isolation

Primary human CD4$^+$ T cells were isolated from leukocyte cones using the RosetteSep Human CD4$^+$ T Cell Enrichment Cocktail (STEMCELL Technologies 15062). T cells were cultured at $37\,°C$ and 5% CO$_2$ in complete RPMI 1640 supplemented with 50 U/mL IL-2 (PeproTech 200-02). Every other day, cells were resuspended in fresh medium at a density of $10^6$/mL. Cas9 ribonucleoproteins (RNPs) were prepared by mixing $8.5\,\mu g$ of TruCut Cas9 protein v2 (ThermoFisher A36498) with 150 pmol of sgRNA mix (Trueguide synthetic gRNAs, ThermoFisher) and Opti-MEM (Gibco 31985062) to a final volume of $5\,\mu L$. The RNPs were incubated for 15 min at RT. Different sgRNA were used for *ADAM10*: GGATTCATCCAGACTCGTGG, CCCCATAAATACGGTCCTCA, AAAATGTGCCACCACGAGTC, GATACCTCTCATATTTACAC. $2 \times 10^6$ freshly isolated T cells were washed three times with Opti-MEM and re-suspended at a density of $2 \times 10^7$/mL. T cells were mixed with the RNPs and transferred into a BTX Cuvette Plus electroporation cuvette (2 mm gap, Harvard Bioscience). The cells were electroporated using a BTX ECM 830 Square Wave Electroporation System (Harvard Bioscience 45-2052) at 300 V, 2 ms, 1 pulse. Immediately following electroporation, cells were transferred to complete RPMI media supplemented with IL-2 at 50 U/mL (PeproTech 200-02). Dynabeads Human T-Activator CD3/CD28 (Thermofisher 11131D) were added after electroporation. 3 and 5 days after activation, 1 mL of medium was exchanged with fresh IL-2 supplemented GM. The beads were magnetically removed on day 6 after electroporation and T cells were re-suspended at $1 \times 10^6$ cells/mL with IL-2 every other day. T cells were maintained until day 17 after electroporation. The knock-out efficiency of the different targets was 40–90%. T cells with residual target protein expression were depleted by antibody staining and bead pull-down or cell sorting (BD Aria III). T cells were re-suspended in MACS buffer (PBS, 0.5% BSA, 2 mM EDTA) at a density of $10^7$/mL. Cells were stained with $5\,\mu L$ of the corresponding PE or APC-labeled antibody per $10^6$ cells for 15 min at $4\,°C$, washed with MACS buffer and re-suspended at a density of $10^8$/mL. $1\,\mu L$ of anti-PE or anti-APC nanobeads (StemCell 17681) were added per $10^6$ cells and incubated on ice for 15 min. After washing with MACS buffer, magnetic beads were added to the samples and incubated for 10 min. After washing, the beads were pulled-down magnetically. The negatively-selected cells were collected and analyzed for surface expression of the target protein by flow cytometry as describe above. Knockout of the molecules was confirmed by SDS-PAGE and western blotting using rabbit anti-ADAM10 polyclonal (Merck AB19026) used at 1/500 followed by goat anti-rabbit HRP (BioRad 1706505) used at 1/1000. Bands were detected using Pierce ECL western blotting substrate (ThermoFisher 32106) and imaged using Hyperfilm (Cytiva Amersham 10084764).

## Lentiviral transduction of CEM-CD43$_{KO}$ with Myc-, Halo- and SNAP-tagged forms of CD43

*CD43* cDNA (NCBI reference sequence: NM_001030288.1) with an N-terminal *Myc*-tag (CD43$_{Myc}$) was purchased from Sino Biological, HG13108-CM. Subsequently, *CD43$_{Myc}$* was cloned into the PEF1 plasmid allowing lentivirus production and encoding puromycin as a selection marker. *CD43$_{Myc}$* lentiviral particles were subsequently produced by polyethyleneimine (PEI) transfection of $1 \times 10^6$ HEK 293 T cells using $2\,\mu g$ p8.91 lentiviral packaging vector, $2\,\mu g$ pMD-G expressing the

VSVG envelope glycoprotein, and 2 μg PEF1 $CD43_{Myc}$. Lentiviral particles produced were then used for transduction of CEM-CD43$_{KO}$ and CEM-ADAM10$_{KO}$ cells. After puromycin selection, CEM-CD43$_{KO}$CD43$_{Myc}$ (termed CD43$_{Myc}$) and CEM-ADAM10$_{KO}$CD43$_{Myc}$ (termed A10$_{KO}$CD43$_{Myc}$) were characterized for expression by flow cytometry using anti-Myc tag 9E10 (Abcam Ab32) at 1/100 with appropriate secondary reagent. $CD43$ cDNA (NCBI reference sequence: NM_001030288.1) with an N-terminal Halo-tag or a C-terminal SNAP tag was kindly provided by Simon Davis. After sub-cloning to obtain both N-terminal Halo and C-terminal SNAP tags in $CD43$ ($CD43_{Halo/SNAP}$), the cDNA was cloned into a lentiviral packaging plasmid and $CD43_{Halo/SNAP}$ lentiviral particles were subsequently produced by polyethyleneimine (PEI) transfection of $1 \times 10^6$ HEK 293 T cells using 2 μg p8.91, 2 μg pMD-G and 2 μg $CD43_{Halo/SNAP}$. Lentiviral particles were then used for transduction of CD43$_{KO}$ CEM and CD43$_{KO}$/ADAM10$_{KO}$ CEM cells. Transduced cells were flow-sorted using a BD FACSAria III for single- and double-positive CD43$_{Halo/SNAP}$ and ADAM10$_{KO}$CD43$_{Halo/SNAP}$ and further characterized by flow cytometry.

### T cell surface biotinylation, apoptosis induction and western blotting

CEM WT and ADAM10 KO cells were biotinylated using the Pierce cell surface biotinylation and isolation kit (ThermoFisher A44390). After biotinylation, apoptosis was induced in half of the samples by incubating with 10 μM Staurosporine, the other control half with vehicle (10 μM DMSO) for 3 h at 37 °C. Samples were centrifuged at $13,000 \times g$ for 10 min to sediment cell pellets and apoptotic vesicles, and cells and supernatant put on ice, and lysis buffer consisting of 1% Triton-X (Sigma-Aldrich 9036-19), 10 nM NEM (Sigma-Aldrich 128-53-0), 1 mM PMSF (Sigma-Aldrich 329-98-6) and 1 x Protease Inhibitor Cocktail (Sigma-Aldrich 539132) was added. Cells were lysed for 30 min at 4 °C with rotation. After removal of cell debris by centrifugation, biotinylated molecules were pulled down using the Pierce Cell Surface Biotinylation and Isolation Kit. Proteins were not removed from the beads as per the kit's instructions, but instead, NuPAGE LDS Sample Buffer (4X) (Thermo Fisher NP0007) was added immediately and the samples boiled at 95 °C for 10 min. The samples were run on a 4–12% gradient gel (Thermo Fisher NP0322PK2) with SeeBlue Plus2 pre-stained protein standard (ThermoFisher LC5925) at 120 V for 90 min. Gels were soaked and transferred onto activated PDVF membranes (Amersham GE10600023), using a semi-dry transfer system (BioRad) at 25 V for 30 min. Membranes were blocked with 5% milk in PBS-Tween for 1 h at RT with rotation. Anti-CD43 clone SP55 (Invitrogen MA5-16339), anti-PSGL-1 clone 108 (Thermo Fisher MA5-29555), anti-MUC1 clone MH1-CT2 (ThermoFisher MA5-11202) and anti-CD45 clone EP322Y (Abcam Ab40763) were all used at 1:1000 in 5% milk in PBS-Tween for 2 h at RT with rotation. After three washes with PBS-Tween, HRP-conjugated secondary antibodies were used at 1:1000 at RT for 1 h with rotation (Anti-rabbit-HRP, Invitrogen 32460, anti-Armenian hamster-HRP, ThermoFisher PA1-32045). Blots were imaged using an Amersham Imager 600, and density quantified using LI-COR ImageStudio V5.2.5 software.

### Multispectral flow cytometry (ImageStream) assay of efferocytosis

MDM were re-plated at $5 \times 10^6$ cells per 6 cm petri dish (Corning) 24 h prior to coculture with T cells. On the day of coculture, T cell lines were labeled with 0.5 μM Cell Tracker Green CMFDA (Invitrogen C2925) and MDM were stained with 1 μM Cell Tracker Orange CMRA (Invitrogen C34551) for 30 min at 37 °C and both were washed with complete RPMI. Primary T cells were not pre-labeled with cell tracker, but pre-treated or not with 10 μM GI or GW for 40 min. T cells were then treated with 10 μM Staurosporine for 3 h at 37 °C to induce apoptosis in about 50% of cells. Cells were washed, stained with near-IR fixable viability

dye, counted and co-cultured with MDM at 1:1 for the indicated time. After removing supernatant, un-attached T cells were removed in cold 10 mM EDTA in PBS for 10 min on ice, and MDM lifted by gentle scraping followed by washing with cold PBS. After fixation with 4% PFA for 10 min at RT, intracellular staining was performed as described above including anti-human active caspase-3 clone Asp175 (Invitrogen PA5-114687) at 1:400, anti-human CD3 clone clone UCHT1-V450 (BD Bioscience 560366) at 1 μg/mL, anti-Myc clone 9E10 (Santa Cruz SC-40) 2 μg/mL, and anti-rabbit IgG (H + L) Alexa Fluor 647, (Invitrogen, A21245) or Alexa Fluor 546 (Invitrogen, A11010), both used at 4 μg/mL (1:500). Data were acquired on an ImageStream MK II (Amnis) with INSPIRE V4 software and analyzed using IDEAS 6.2 software. Gating on the relevant cell population was set according to in-focus images before doublet exclusion. The Cell Tracker Green and Orange double labeled population was sub-divided into engulfed, attached and unattached T cells based on the delta-centroid function, which measures the distance between the center of MDM and T cells within each image. Subsequent gating was carried out on active caspase-3 labeling to differentiate apoptotic from non-apoptotic cells within the engulfed population. The engulfment of specific T cell subsets of non-apoptotic compared to apoptotic cells was used to generate an uptake index, which is the [ratio of % MDM-engulfed apoptotic T cells /% apoptotic T cells prior to co-culture] x 100. A similar index was generated for non-apoptotic cells using [ratio of % MDM-engulfed non-apoptotic cells /% non-apoptotic T cells prior to coculture] x 100.

### Validation of the efferocytosis assay

To confirm that apoptotic T cells were being engulfed by MDM as quantified by the Imagestream assay, MDM were incubated with or without previously optimized concentrations of 1 μM cytochalasin D (Sigma-Aldrich C8273), 1 μM jasplakinolide (Abcam ab141409) or 1 μM latrunculin A (Abcam ab144290) for 1 h at 37 °C prior to coculture with apoptotic T cells carried out as described above, and inhibitors remained present throughout the coculture. After T cell coculture MDM were put in cold 10 mM EDTA in PBS for 10 min on ice, gently scraped and washed with cold PBS, stained with near-IR fixable viability dye (Invitrogen L34994) diluted 1:1000, and cell viability quantified by flow cytometry. To assess actin remodeling inhibitor effects on MDM metabolic activity, MDM treated with or without actin remodeling inhibitors as above were washed and incubated with 50 μL serum-free media and 50 μL of MTT reagent (Abcam MTT assay, ab211091) per well, and incubated at 37 °C for 3 h. 150 μL of MTT solvent was added into each well after the incubation, the plate was gently shaken for 15 min in the dark and absorbance read at OD = 590 nm on a SpectraMax M5 plate reader (Molecular Devices). To ensure that the actin remodeling inhibitors blocked phagocytosis, MDM treated or not as described above were cultured with latex bead-rabbit IgG-FITC complexes (Cayman Chemical phagocytosis assay kit 500290) at 1:300 for 1 h at 37 °C. After washing off the supernatant, MDM were put in cold 10 mM EDTA in PBS for 10 min on ice, gently scraped and washed with cold PBS. Surface FITC fluorescence was quenched to distinguish internalized beads from surface attached beads using trypan blue quenching solution for 2 min, followed by a wash with assay buffer. MDM were fixed with 4% paraformaldehyde (Sigma Aldrich) for 10 min at RT and analyzed by flow cytometry.

### Fluorescence correlation spectroscopy and fluorescence cross-correlation spectroscopy

To prepare the cells and supernatants for fluorescence correlation spectroscopy (FCS) and fluorescence cross-correlation spectroscopy (FCCS) measurements[56], CEM-CD43$_{Halo/SNAP}$ cells with or without ADAM10$_{KO}$ ($2.5 \times 10^5$ cells per condition) were washed twice and

resuspended in complete RPMI without antibiotics. All washing and centrifugation steps were carried out at 1500 rpm for 1 min. HaloTag-AlexaFluor 488 Ligand (Promega G1001) for labeling of the CD43 ectodomain was added to the cells at a final concentration of 0.33 μM, gently mixed and incubated at 37 °C for 1 h. For FCCS experiments and exclusion of CD43 loss in apoptotic bodies, CD43$_{Halo/SNAP}$ cells were additionally labeled at the C-terminus with SiR650-BG SNAP-tag substrate (Spirochrome AG SC504) at a final concentration of 0.33 μM at 37 °C for 30 min. After labeling, $2.5 \times 10^5$ cells per condition were washed twice and resuspended in 500 μL of un-supplemented phenol red-free L15 medium (ThermoFisher 21083027). Cells were left untreated, treated with 10 μM staurosporine at 37 °C for 3 h, or pre-treated with the inhibitor GI (1:1000 dilution) at 37 °C for 45 min and subsequently treated with 10 μM staurosporine at 37 °C for 3 h. Thereby, the total time until further processing was kept constant for all conditions within one experiment. The cells were pelleted, supernatant centrifuged and transferred onto a pre-washed 0.5 ml 40 K MWCO Zeba spin desalting column (ThermoFisher A57756). The flow-through was further clarified twice using 40 K columns and the final flow-through analyzed by FCS. For FCCS measurements, $2.5 \times 10^5$ cells were Halo- & SNAP-labeled as above, washed once in fully supplemented RPMI, resuspended and incubated in RPMI at 37 °C for 30 min and then washed twice and resuspended in 500 μl L15 medium. 100 μl of cell suspension was transferred into an μ-slide 18-well glass bottom (Ibidi 81817). FCS measurements were carried out using Zeiss LSM 980 microscope. 488 nm argon ion and 633 nm He-Ne lasers were used to excite A488 and SiR, respectively, associated with a 40x/1.2 NA water immersion objective. 10 curves (10 s each) were taken per sample. Laser power was set to 1% of the total laser power that corresponds to ≈10 μW. Intensity traces were used to count the peaks. To obtain the diffusion coefficients, curves were fitted using the 3D-diffusion model below to obtain the transit time (Eq. 1).

$$G(\tau) = \frac{1}{N} \left(1 + \frac{\tau}{\tau_D}\right)^{-1} \frac{1}{\sqrt{1 + K^2 \frac{\tau}{\tau_D}}} \quad (1)$$

where $\tau_D$ represents the transit time through the observation spot. The diffusion coefficients were calculated as follows (Eq. 2).

$$\tau_D = \frac{\omega^2}{8\ln(2)D} \quad (2)$$

Where $w$ corresponds to the full-width half maximum of the point spread function; $\tau_D$ is the diffusion time and $D$ is diffusion coefficient.

## Statistical analysis and reproducibility

All group sizes are reported in the figure legends where n = individual experiments and/or individual donors. In some experiments, different group sizes were compared resulting in a range of n. No technical replicates are included in the data presentation or analysis. In some experiments (Fig. 1c, d; Fig. 2b, i–k; Fig. 3c) data were normalized to mucin expression on non-apoptotic (Non-apop) cells set at 100% and shown as a single black bar. All data sets were tested where appropriate for normal distribution using the Kolmogorov–Smirnov test. Where data were normally distributed, Student's $t$-test for two-way comparisons, paired or unpaired, or ANOVA for multiple comparisons were applied with Dunnet's post-test correction. Where data did not conform to a normal distribution they were either $\log_{10}$-transformed where appropriate and subjected to ANOVA for multiple comparisons with Dunnet's post-hoc test, or subjected to Mann–Whitney $U$ for two-way comparisons or Kruskal–Wallis for multiple comparisons with Dunn's post-hoc test. All error bars are represented as ±1 standard deviation (SD) around the mean.

## Reporting summary

Further information on research design is available in the Nature Portfolio Reporting Summary linked to this article.

## Data availability

All data are either reported in the manuscript or supplied as separate source data files provided with this paper or from the author on request. No custom code has been used in this paper, and mathematical algorithms are reported in the methods section. Reagents are available where applicable through an institutional MTA agreement. Source data are provided with this paper.

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

## Acknowledgements

The authors thank current and past members of the Sattentau lab and the Sir William Dunn School of Pathology, Michael D Moore for helpful discussions concerning the potential cleavage of CD43, and Daniel Fletcher and Emily Suter for discussion and advice during the design and implementation of the project. We thank Simon Davis for generously providing Halo- and SNAP-tagged CD43 constructs, Linda Baum for supplying the CEM-CD43 knock-out cells, Eric Rubenstein for kindly supplying 11G2 antibody, and Yoshi Itoh for initial supply of ADAM10 inhibitors. We thank Robert Hedley for core flow cytometry and Imagestream assistance, Alan Wainman for imaging assistance, Jesús Siller Farfán for assistance with MESF quantification, and Alun Vaughan-Jackson for advice on performing CRISPR KO. Funding is acknowledged as follows: MR/S009329/1 from the UKRI Medical Research Council (Q.J.S., N.K.), China Scholarship Council-University of Oxford Scholarship 201908130151 (S.Z.), Wellcome Trust Studentship 108869/Z/15/Z (L.Z.D.), Wellcome Trust Senior Fellowship in Basic Biomedical Sciences (207537/Z/17/Z) (O.D.), UKRI-Biotechnology and Biological Sciences

Research Council BB/T008784/1 (J.C.C.), Karolinska Institutet, SciLife-Lab, Swedish Research Council Starting Grant (2020-02682) (E.S., F.R.), SciLifeLab National COVID-19 Research Program financed by the Knut and Alice Wallenberg Foundation, Cancer Research KI and Human Frontier Science Program (RGP0025/2022) (E.S., F.R.).

## Author contributions

Conceptualization of initial project: N.K., Q.J.S.; additional conceptualization: L.Z.D., S.Z. Methodology: N.K., L.Z.D., S.Z., M.D.P., R.A.R., O.D., E.S., Q.J.S. Investigation: L.Z.D., S.Z., M.D.P., F.R., E.S., L.P.D., J.C.C., H.R., R.A.R., E.S., and N.K. Funding acquisition: Q.J.S., N.K. Project administration: Q.J.S. Supervision: Q.J.S., L.Z.D., M.D.P., R.A.R., O.D., E.S. Writing – original draft: Q.J.S., L.Z.D., S.Z., F.R., E.S. Writing – review & editing: L.Z.D., S.Z., M.D.P., F.R., E.S., L.P.D., J.C.C., H.R., R.A.R., O.D., E.S., N.K., and Q.J.S.

## Competing interests

The authors declare no competing interests.
