## [Peer Review File · Nature Communications]

Apoptosis-mediated ADAM10 activation removes a mucin barrier promoting T cell efferocytosisREVIEWER COMMENTS

Reviewer #1 (Remarks to the Author):

The authors demonstrate that apoptotic T cells reduce their “don’t eat me” barrier of mucins (CD43, PSGL-1 and MUC-1) via their shedding by the metalloprotease ADAM10. The latter is shown to be activated via caspase 3-mediated activation of the scramblase XKR8, which flips negatively charged phosphatidylserine from the inner to the outer leaflet of the plasma membrane, which can then engage with a positively charged region on ADAM10 leading to enhanced shedding activity. As a consequence of these events, apoptotic T cell phagocytosis by macrophages (termed efferocytosis) is increased.

This is original work that is of high significance for the following reasons:

- Efferocytosis of apoptotic T cells is central to the function of the immune system in a variety of processes to maintain health, e.g. development and tissue homeostasis and repair. Furthermore, its dysregulation is associated with human diseases such as infection, cancer, inflammatory and autoimmune diseases.
- ADAM10 is of major interest because it is essential for organismal development as the critical activator of Notch cell fate regulators. It is also implicated in a variety of diseases such as cancer, inflammatory and neurodegenerative diseases. However, the importance of ADAM10 in shedding non-Notch substrates remains relatively poorly characterised, and more work in this area is required if ADAM10’s potential as a drug target is to be realised.
- The study strengthens the case made by a recent publication that ADAM10 can be activated by phosphatidylserine exposure on the outer leaflet of the plasma membrane, and extends this to apoptosis-induced activation of the scramblase XKR8.

The study uses an impressive range of methodology and the data presentation is generally of a high standard using appropriate controls, many included in the ten supplementary figures that accompany the four main figures. The inclusion of some model figures to explain experiments is helpful. The authors acknowledge that the work is in vitro using a human T cell line and primary human T cells and macrophages. However, due to the impressive mechanistic work in the study and its high significance, in vivo work is not warranted, in my opinion. Instead, it will be interesting to follow up the in vivo importance

through future studies.

My suggestions for improvement are as follows.

1. It would be helpful to include the basal expression levels of the mucins following ADAM10 knockout in the CEM T cell line and primary T cells. The western blotting data in Figure 2F-G suggest that basal levels may be elevated, as would be predicted, but it would be useful to show this via flow cytometry with quantitation.
2. In Figure 2A, the panel labels at the top are missing.
3. In Supplementary Figure 4A and D, could the authors comment on why the knockout of ADAM10, as measured by flow cytometry, is not 100%?
4. In the Methods on line 367, is the statement that “XKR8 is an intracellular molecule” incorrect? My understanding, and Figures 3A and D and Supplementary Figure 10, is that it can be plasma membrane-localised.
5. There is increasing evidence that ADAM10 exists in an intimate heterodimer complex with one of six different regulatory tetraspanins, e.g. see Koo 2020 J Biol Chem and Lipper 2022 bioRxiv. It would be useful to the reader to point this out, perhaps in an extra couple of sentences in the Discussion, which could include speculation that one or more specific tetraspanin/ADAM10 complexes could be responsible for mucin shedding.

Reviewer #2 (Remarks to the Author):

In this work, Drexhage, Sattentau, and colleagues present a series of data that suggest that ADAM-10 mediated cleavage of mucin molecules such as CD43, PSGL1 and MUC1 may contribute to their 'clearance' proximal to the exposed PS and thereby facilitate efferocytosis by macrophages. They also suggest based on knockout cell lines that Xkr8 scramblase-mediated PtdSer exposure may activate the ADAM10 (based on evidence in the literature on the activation of ADAM10 by PtdSer), and provide a link to these processes.

Overall, this is a very well done manuscript, and the data support the conclusions. I have only have a few minor comments, which can all be addressed via text changes rather than new experiments.

1. While the data from Xkr8 knockout support the notion that PS exposure links to mucin cleavage, this is not direct evidence. Despite the lactadherin data that the authors provide, it is still correlative as the authors did not directly demonstrate in this work that the apoptotic exposed PS is responsible for ADAM10 activation. Therefore, the authors should soften this conclusion both in the main text and in the discussion.

2. Although not required, another experiment that the authors may consider doing is to test whether calcium ionophore-induced and TMEM16F-dependent PS exposure would also lead to ADAM10 activation and mucin shedding, or the ADAM10 activation is ONLY dependent on apoptotic PS exposure. This could have significant implications for PS exposure on live vs apoptotic cells.

3. Some of the comments in the discussion about cancer cells and their uptake assumes that recognition of cancer cells in the tumor by macrophages is analogous to the apoptotic PS exposure and mucin cleavage that the authors see, while this may not be the case in live cells. Further, ADAM10 has many targets and at this point we really do not know the plethora of proteins that may be cleaved in different circumstances. Therefore, this is a bit of a stretch. The authors have already demonstrated a nice role in apoptosis and this need not be extended to live cells or cancer cells until further data is obtained.

Reviewer #3 (Remarks to the Author):

Drexhage and colleagues investigated the removal of mucins during apoptosis by ADAM10 and the importance of this process in efferocytosis. Interestingly, shedding of mucins by ADAM10 is regulated by the flippase XKR8. Overall, the manuscript is novel and highly relevant to the field of cell clearance. The experiments presented were well performed and the conclusions were largely justified. In particular, the demonstration that mucins are removed by ADAM10 during apoptosis is convincing. The authors should, however, consider the following points:

1. Did lactadherin treatment affected AV binding in the flow cytometry assay and thus affect the quantification of the percentage of apoptosis?

2. Did the authors validate the ImageStream-based engulfment assay with controls like cyto-D treatment to block engulfment?

3. Figure 4d: I am a bit unsure regarding the interpretation of the analysis, as the data demonstrate CD43myc overexpressing cells are not well engulfed rather than selective uptake of CD43myc depleted cells? Can the author please clarify.

4. Figure 4c and 4e: As the hypothesis of the study is that mucins like CD43, PSGL and MUC1 are lost during apoptosis, and the loss of these mucins aid efferocytosis. Can the authors please comment on the need to generate KO of these mucins for the efferocytosis assay? Is this to test the residual effects of some uncleaved CD43, PSGL and MUC1 on apoptotic cells?

5. Can the author please comment on the role of mucin on the initial interaction between the phagocyte and dying cells (e.g. from ImageStream analysis of attachment)?

6. The data presented is convincing enough and the following comment is beyond the scope of the current study, but can the basic patch on ADAM10 be mutated or is that not feasible?

Minor points:

Page 4: Labelling error for Figure 1h. It should be Figure 1e?

Figure 2a: Please label the panel clearly regarding the cell surface molecules being monitored. Annotation similar to Figure 2b would be helpful for the reader.

Supplementary Figure 6c: The '+' and '-' is a bit confusing next to the flow cytometry histograms, simply indicate KO versus WT for XKR8 maybe clearer?

Can the authors comment on the role of mucins in the removal of necroptotic and pyroptotic cells? If appropriate, can include a sentence in the discussion.

Reviewer #4 (Remarks to the Author):

The removal of cells undergoing apoptosis by phagocytes is an important process within the apoptosis pathway, safeguarding against secondary necrosis of apoptotic cells leading to expulsion of cellular constituents that drive an inflammatory response. While a great deal is known about how phagocytes recognise cells undergoing apoptosis by interacting with eat-me-signals on the dying cells using dedicated receptors and how engulfment of cells by phagocytes can be hindered through don't eat me signals, there is still more to be learnt about the process of engulfment of dying cells.

In this manuscript Linnea Drexhage and colleagues present data showing that mucins on the surface of cells undergoing apoptosis hinder uptake of cells undergoing apoptosis, that these mucins are shed from dying cells by a process dependent on the metallo-protease ADAM10 which in turn is activated by XKR8 dependent re-localisation of phosphatidyl-serine (PS) from the inner side of the plasma membrane to the outer side in cells undergoing apoptosis. The data shown are of very good quality and interesting. My major concern is that there is insufficient depth in data in this paper in this current form. However, I believe that if the authors will conduct several more investigations, a substantially revised paper could be suitable for publication in Nature Communications.

Figure 1: Not only one T lymphoma line but at least 3 should be examined for mucin expression.

Out of interest, the authors should also test some B lymphoma lines, some AML cell lines and some solid cancer derived cell lines for mucin expression. This would allow generalization of the findings presented.

Figure 1: Staurosporine and UV radiation are not physiological inducers of apoptosis for T cells. The authors should also treat cells with glucocorticoids and subject them to nutrient deprivation to induce apoptosis.

Importantly, it would also be very interesting to know if mucin levels on the surface of cells will also drop when they are stimulated to undergo necroptosis or pyroptosis, or whether the process described here by the authors is specific to apoptosis.

Supplementary Figure 1: Stimulation with PHA+ionomycin is not a physiological activation

of human T cells. The authors should also activate primary human T cells with mitogenic antibodies against CD3 and CD28 and then test changes in the surface expression of mucins.

Supplementary Figure 2: This experiment should be done also with at least one physiological inducer of apoptosis, such as nutrient deprivation or treatment with glucocorticoids.

Also the experiment should be conducted with stimuli that induce necroptosis or pyroptosis to examine whether the changes in mucin levels on the cell surface are restricted to apoptosis or can also occur when cells undergo different forms of programmed cell death.

Figure 2a-e: These experiments should be done also with at least one physiological inducer of apoptosis, such as nutrient deprivation or treatment with glucocorticoids.

Also, the experiment should be conducted with stimuli that induce necroptosis or pyroptosis to examine whether the changes in mucin levels on the cell surface are restricted to apoptosis or can also occur when cells undergo different forms of programmed cell death.

Supplementary Figure 4: These experiments should be done also with at least one physiological inducer of apoptosis, such as nutrient deprivation or treatment with glucocorticoids.

Also, the experiment should be conducted with stimuli that induce necroptosis or pyroptosis to examine whether the changes in mucin levels on the cell surface are restricted to apoptosis or can also occur when cells undergo different forms of programmed cell death.

Supplementary Figure 5: These experiments should be done also with at least one physiological inducer of apoptosis, such as nutrient deprivation or treatment with glucocorticoids.

Also, the experiment should be conducted with stimuli that induce necroptosis or pyroptosis to examine whether the changes in mucin levels on the cell surface are restricted to apoptosis or can also occur when cells undergo different forms of programmed cell death.

Figure 2f-j: These experiments should be done also with at least one physiological inducer of apoptosis, such as nutrient deprivation or treatment with glucocorticoids.

Also, the experiment should be conducted with stimuli that induce necroptosis or pyroptosis to examine whether the changes in mucin levels on the cell surface are restricted to apoptosis or can also occur when cells undergo different forms of programmed cell death.

Figure 3: These experiments should be done also with at least one physiological inducer of apoptosis, such as nutrient deprivation or treatment with glucocorticoids.

Also, the experiment should be conducted with stimuli that induce necroptosis or pyroptosis to examine whether the changes in mucin levels on the cell surface are restricted to apoptosis or can also occur when cells undergo different forms of programmed cell death.

Figure 4: These experiments should be done also with at least one physiological inducer of apoptosis, such as nutrient deprivation or treatment with glucocorticoids.

Also, the experiment should be conducted with stimuli that induce necroptosis or pyroptosis to examine whether the changes in mucin levels on the cell surface are restricted to apoptosis or can also occur when cells undergo different forms of programmed cell death.

Figure 4: ADAM10 should be knocked out not only in CEM cells but also in at least one additional T lymphoma cell line to allow generalisation of the findings presented.

All Western blots presented need to show molecular weight markers.

REPLIES TO REVIEWER COMMENTS. NB, reviewer comments in black, author replies in blue.

Reviewer #1 (Remarks to the Author):

The study uses an impressive range of methodology and the data presentation is generally of a high standard using appropriate controls, many included in the ten supplementary figures that accompany the four main figures. The inclusion of some model figures to explain experiments is helpful. The authors acknowledge that the work is in vitro using a human T cell line and primary human T cells and macrophages. However, due to the impressive mechanistic work in the study and its high significance, in vivo work is not warranted, in my opinion. Instead, it will be interesting to follow up the in vivo importance through future studies.

We thank the reviewer for their thoughtful and positive summary of the work and for the helpful conclusion that in vivo work is not warranted.

My suggestions for improvement are as follows.

1. It would be helpful to include the basal expression levels of the mucins following ADAM10 knockout in the CEM T cell line and primary T cells. The western blotting data in Figure 2F-G suggest that basal levels may be elevated, as would be predicted, but it would be useful to show this via flow cytometry with quantitation. This is a great idea and we now include these data as new Supplementary Fig. 5f, g and in the revised manuscript in lines 143-151. As the reviewer surmised, in CEM-A10_{KO} we observe a general trend towards increased mucin expression compared to CEM_{WT} but which reaches significance only for PSGL-1 ($p < 0.05$, Supplementary Fig. 5f). In primary A10_{KO} CD4⁺ T cells we observed a similar trend towards modest increases in basal levels of the mucins, with only CD43, and, interestingly, MUC24, showing significance ($p < 0.05$, Supplementary Fig. 5g). We assume that these effects result from very low level constitutive ADAM10 shedding in the healthy WT cells, which we interpret (new lines 150-151) as implying tight regulation of basal ADAM10-mediated mucin cleavage as would be predicted from exclusion of PS on the outer plasma membrane leaflet of healthy cells.

2. In Figure 2A, the panel labels at the top are missing. Thank you this is now corrected.

3. In Supplementary Figure 4A and D, could the authors comment on why the knockout of ADAM10, as measured by flow cytometry, is not 100%? We thank the reviewer for picking this up. This resulted from the use of an incorrect isotope control antibody to represent background labelling. We have now included the correct isotope control and show that the ADAM10 signal in the KO cells is essentially background. New Supplementary Figs. 5a and 5d have been revised accordingly.

4. In the Methods on line 367, is the statement that "XKR8 is an intracellular molecule" incorrect? My understanding, and Figures 3A and D and Supplementary Figure 10, is that it can be plasma membrane-localised. The reviewer is indeed correct that XKR8 has extracellular loops, this was an error on our part which we have corrected in the revised manuscript methods section line 434. Nevertheless, the point stands that there are no validated antibodies specific for XKR8 and so we were unable to screen for protein expression on the plasma membrane by flow cytometry or total protein by Western blotting, as highlighted in lines 434 and 438-440.

5. There is increasing evidence that ADAM10 exists in an intimate heterodimer complex with one of six different regulatory tetraspanins, e.g. see Koo 2020 J Biol Chem and Lipper 2022 bioRxiv. It would be useful to the reader to point this out, perhaps in an extra couple of sentences in the Discussion, which could include speculation that one or more specific tetraspanin/ADAM10 complexes could be responsible for mucin shedding. We thank the reviewer for raising this good point, and have now added a brief section to the revised discussion (lines 275-279) in which we consider how tetraspanins might regulate ADAM10 function in the current context, citing two new references (Harrison et al 2021, PMID: 34201472; new ref 39, and Koo et al 2020, PMID: 32111735; new ref 40) in this respect. This has been of particular relevance in light of the new results we have obtained in response to the proposal by reviewer 4 to test different immortalized leukocyte cell lines for mucin expression and loss. The selectivity of mucin cleavage by ADAM10 in the monocyte line differs somewhat from T cells, implying that ADAM10 substrate specificity is cell type-specific and likely to be modulated by association with tetraspanins.

Reviewer #2 (Remarks to the Author):

In this work, Drexhage, Sattentau, and colleagues present a series of data that suggest that ADAM-10 mediated cleavage of mucin molecules such as CD43, PSGL1 and MUC1 may contribute to their 'clearance' proximal to the exposed PS and thereby facilitate efferocytosis by macrophages. They also suggest based on knockout cell lines that Xkr8 scramblase-mediated PtdSer exposure may activate the ADAM10 (based on evidence in the literature on the activation of ADAM10 by PtdSer), and provide a link to these processes. Overall, this is a very well done manuscript, and the data support the conclusions. I have only have a few minor comments, which can all be addressed via text changes rather than new experiments.

We thank the reviewer for their clear summary of the work and their positive and supportive comments regarding the study.

1. While the data from Xkr8 knockout support the notion that PS exposure links to mucin cleavage, this is not direct evidence. Despite the lactadherin data that the authors provide, it is still correlative as the authors did not directly demonstrate in this work that the apoptotic exposed PS is responsible for ADAM10 activation. Therefore, the authors should soften this conclusion both in the main text and in the discussion. We agree with the reviewer's point and have now nuanced our consideration of the data regarding the role of XKR8 in ADAM10 activation in lines 32–34, 181, 192-195, 199 and 267-268 of the revised manuscript, and present the data as correlative rather than mechanistically definitive.

2. Although not required, another experiment that the authors may consider doing is to test whether calcium ionophore-induced and TMEM16F-dependent PS exposure would also lead to ADAM10 activation and mucin shedding, or the ADAM10 activation is ONLY dependent on apoptotic PS exposure. This could have significant implications for PS exposure on live vs apoptotic cells. We agree with the reviewer that this is a very interesting logical progression of the work linked to PS flipping and ADAM10 activity in relation to T cell activation. We have attempted some preliminary experiments in this regard, and although we do see some mucin downregulation linked to ionophore-induced T cell activation, the data are complex and we would need to dedicate substantial time and effort to clarify the results and provide appropriate controls and context. Since the current manuscript focusses on apoptosis and efferocytosis, and the reviewer states that this aspect of the revision is not required, we hope that they will understand that we have chosen not to pursue this body of work in the current manuscript. However, in response to the reviewer's comment we do now discuss a possible role for TMEM16F in calcium-mediated PS flipping via TMEM16F, new lines 282–286, and have added new references 41-43 in this regard.

3. Some of the comments in the discussion about cancer cells and their uptake assumes that recognition of cancer cells in the tumor by macrophages is analogous to the apoptotic PS exposure and mucin cleavage that the authors see, while this may not be the case in live cells. Further, ADAM10 has many targets and at this point we really do not know the plethora of proteins that may be cleaved in different circumstances. Therefore, this is a bit of a stretch. The authors have already demonstrated a nice role in apoptosis and this need not be extended to live cells or cancer cells until further data is obtained. We thank the reviewer for this thoughtful suggestion and agree that we had perhaps speculated a little too freely in terms of implications for cancer cells and non-apoptotic cells. We have now removed the sentences (previous lines 244–249) in the discussion concerning live cells and cancer cells in the revised manuscript, and focus instead on discussing the role of mucins during cell death and efferocytosis.

Reviewer #3 (Remarks to the Author):

Drexhage and colleagues investigated the removal of mucins during apoptosis by ADAM10 and the importance of this process in efferocytosis. Interestingly, shedding of mucins by ADAM10 is regulated by the flippase XKR8. Overall, the manuscript is novel and highly relevant to the field of cell clearance. The experiments presented were well performed and the conclusions were largely justified. In particular, the demonstration that mucins are removed by ADAM10 during apoptosis is convincing. The authors should, however, consider the following points: We thank the reviewer for their supportive and positive assessment of the work.

1. Did lactadherin treatment affected AV binding in the flow cytometry assay and thus affect the quantification of the percentage of apoptosis?

The reviewer makes a pertinent point. Indeed, the positive control for the assay was based on lactadherin blocking annexin-V binding as the reviewer suggests. However, in these assays apoptotic T cells were gated using detection of cleaved caspase-3, and so the lactadherin treatment would not have influenced apoptosis quantification. The caspase-3 gating for apoptotic cells was mentioned in the main text (original line 190) and legend to original Fig. 3e, but for clarity we have now also introduced a separate methods section to better describe how we did this experiment (new lines 407-412) and also mention the use of caspase-3 labelling in the legend to new Supplementary Fig. 7e.

2. Did the authors validate the ImageStream-based engulfment assay with controls like cyto-D treatment to block engulfment?

This is an excellent point, and we fully agree this is an essential control to include. We now supply additional methods and data showing that inhibition of macrophage actin cytoskeleton remodelling by non-toxic concentrations of the actin depolymerising agents cytochalasin D and latrunculin, and the actin stabilising agent jasplakinolide, prevents apoptotic T cell uptake by macrophages in the Imagestream assay. The data are supported by new results showing that these actin remodelling inhibitors inhibit uptake of beads in a phagocytosis assay, but are neither cytotoxic nor reduce MDM metabolic activity in an MTT assay. These new data are now presented in new Supplementary Fig. 8c-f, discussed in new lines 211–215, and the methods underlying these experiments presented in new lines 552-572.

3. Figure 4d: I am a bit unsure regarding the interpretation of the analysis, as the data demonstrate CD43myc

overexpressing cells are not well engulfed rather than selective uptake of CD43myc depleted cells? Can the author please clarify.

We agree with the reviewer that this could have been more clearly described. Since apoptotic cells over-expressing CD43^{Myc} are more strongly represented in the starting culture compared to those engulfed by the macrophages, we assume that macrophages are “deselecting” the strongly expressing CD43^{Myc} cells in favour of the CD43^{Myc}-low/negative cells for uptake from the starting culture. We have now improved the presentation of data in new Fig. 4e and make what we believe is a clearer description of the results in the revised text lines 231-234.

4. Figure 4c and 4e: As the hypothesis of the study is that mucins like CD43, PSGL and MUC1 are lost during apoptosis, and the loss of these mucins aid efferocytosis. Can the authors please comment on the need to generate KO of these mucins for the efferocytosis assay? Is this to test the residual effects of some uncleaved CD43, PSGL and MUC1 on apoptotic cells?

We thank the reviewer for pointing out that the rationale for these experiments was not fully explained. We now clarify in the revised manuscript that although these mucins are shed over time during apoptosis, this takes up to 6 hours for maximum mucin loss to occur, and as the reviewer suggests, mucin loss is not complete (Fig. 1a, b). Therefore, if mucins are indeed providing a biophysical barrier to apoptotic T cell uptake, then their absence at the start of the coculture assay should accelerate ‘eat-me’ signal recognition and subsequent attachment and uptake by macrophages. This is indeed what we observe as apoptotic CEM-CD43^{KO} (Fig. 4c) and PSGL-1^{KO} and MUC1^{KO} lines (Fig. 4f) are taken up 1.7-, 2- and 1.9-fold better after 1 h coculture with macrophages than their wild-type counterparts. The rationale behind this experiment is now more clearly explained in new lines 221-229 and 234-235.

5. Can the author please comment on the role of mucin on the initial interaction between the phagocyte and dying cells (e.g. from ImageStream analysis of attachment)?

We thank the reviewer for making this interesting and valid point, as one might indeed expect loss of mucins to increase initial attachment of apoptotic T cells to macrophages prior to engulfment. To interrogate this we have quantified T cell-macrophage attachment data obtained from ImageStream analysis, and observe that attachment of apoptotic CEM-CD43^{KO} T cells to macrophages is significantly increased compared to their WT CEM counterparts. These new data are added as new Supplementary Fig. 9a and considered in revised text lines 224–227.

6. The data presented is convincing enough and the following comment is beyond the scope of the current study, but can the basic patch on ADAM10 be mutated or is that not feasible? We agree with the reviewer that this would have been a nice additional control, and we have attempted this mutagenesis as part of the project. However, we have had substantial difficulty expressing both wild-type and mutated exogenous human ADAM10 in human T cells, and conclude that reporting on this aspect will have to await a subsequent study.

Minor points:

Page 4: Labelling error for Figure 1h. It should be Figure 1e?

Thank you, now corrected in the revised text line 116.

Figure 2a: Please label the panel clearly regarding the cell surface molecules being monitored. Annotation similar to Figure 2b would be helpful for the reader.

Agreed, done.

Supplementary Figure 6c: The ‘+’ and ‘–’ is a bit confusing next to the flow cytometry histograms, simply indicate KO versus WT for XKR8 maybe clearer?

We agree and have implemented this in new Supplementary Fig. 7c.

Can the authors comment on the role of mucins in the removal of necroptotic and pyroptotic cells? If appropriate, can include a sentence in the discussion.

This is definitely of interest, although we consider that it falls outside of the scope of the current study which is focussed on mucin loss and efferocytosis of apoptotic cells. However, as suggested we now mention (new lines 279-282) that other forms of regulated cell death result in phosphatidylserine exposure on the outer leaflet of the plasma membrane and so may well trigger ADAM10 activation and mucin cleavage. This is indeed the subject of an ongoing study.

Reviewer #4 (Remarks to the Author):

In this manuscript Linnea Drexhage and colleagues present data showing that mucins on the surface of cells undergoing apoptosis hinder uptake of cells undergoing apoptosis, that these mucins are shed from dying cells by a process dependent on the metallo-protease ADAM10 which in turn is activated by XKR8 dependent re-localisation of phosphatidyl-serine (PS) from the inner side of the plasma membrane to the outer side in cells

undergoing apoptosis. The data shown are of very good quality and interesting. My major concern is that there is insufficient depth in data in this paper in this current form. However, I believe that if the authors will conduct several more investigations, a substantially revised paper could be suitable for publication in Nature Communications. We thank the reviewer for their careful and comprehensive analysis of the paper and for their supportive comments regarding the interest and quality of the data. We note that this reviewer suggests that there is “insufficient depth in the data”, which is in contrast with the remarks of the other 3 reviewers who all concur (in summary) that the data support the conclusions.

Figure 1: Not only one T lymphoma line but at least 3 should be examined for mucin expression.

Out of interest, the authors should also test some B lymphoma lines, some AML cell lines and some solid cancer derived cell lines for mucin expression. This would allow generalization of the findings presented.

We thank the reviewer for this proposal. We would like to point out that the explicit focus of this study is T cells and their removal by macrophages during efferocytosis, which is why we restricted ourselves to their analysis. We also point out that in the original study we had not only analyzed CEM, but also the Jurkat T-ALL line (original Supplementary Fig. 1f, now new Supplementary Fig. 2c, h) and of course primary CD4⁺ T cells. We therefore consider that the study was comprehensive in terms of depth of T cell analysis, and this is in line with the comments of the other reviewers. However, we agree that studying more cell types has some merit in generalising the phenomenon, and have now analyzed mucin expression on another leukaemic T cell line (HPBALL, new supplementary Fig. 2d) and on primary CD8⁺ T cells (new supplementary Fig. 2a), which both show the same pattern of mucin downregulation during apoptosis as CEM, Jurkat and primary CD4⁺ T cells. At the request of the reviewer we have gone further and also interrogated the expression of mucins on a B cell line (NALM6, new Supplementary Fig. 2e) and a myeloid line (U937, new Supplementary Fig. 2f) and again observe selective mucin loss during apoptosis, although the pattern of basal mucin expression and loss differs somewhat from T cells. These data are integrated in new lines 83-91 and discussed in 273-279. Finally, we have tested the sensitivity of mucin loss on Jurkat, HPBALL, NALM6 and U937 to inhibition by the ADAM10/17 and ADAM10-specific protease inhibitors GW and GI respectively, and confirm that mucin loss on all these lines is sensitive to these inhibitors (new Supplementary Figs. 4f - i), results integrated into new lines 134-136. This leads us to conclude that whilst ADAM10-mediated mucin loss during apoptosis is generalizable to other immune cell types, mucin basal expression and shedding is cell type-specific. Since reviewer 2 requested that we remove discussion of the specific relevance of our observations to cancer cells from the manuscript, with which we concur, we consider that additional work on tumour lines is unwarranted.

Figure 1: Staurosporine and UV radiation are not physiological inducers of apoptosis for T cells. The authors should also treat cells with glucocorticoids and subject them to nutrient deprivation to induce apoptosis.

We agree that apoptosis induction by staurosporine, whilst widely used and a well-accepted model system, has limited physiological relevance. By contrast UV light is a biologically-relevant apoptosis inducer, at least in epithelial cells, which yields data coordinate with our staurosporine results in T cells. Nevertheless, we have followed the advice of the reviewer and used the glucocorticoid dexamethasone, and find that apoptosis induction by this pathway in both T cell lines CEM and Jurkat (Supplementary Fig. 2g, h respectively) and primary CD4⁺ T cells (new Supplementary Fig. 2i) leads to exactly the same pattern of mucin loss. These new data which are now discussed in new lines 91-94 do not change the overall conclusions but make a helpful addition to the manuscript, and we have added the new methods to lines 342-344 and a new reference (25) for dexamethasone induction of apoptosis.

Importantly, it would also be very interesting to know if mucin levels on the surface of cells will also drop when they are stimulated to undergo necroptosis or pyroptosis, or whether the process described here by the authors is specific to apoptosis.

We agree with the reviewer that this is an interesting question, also raised by reviewer 3 (who did not request any additional experimentation). Since the current study is entirely focussed on efferocytosis, which is by definition the clearance of apoptotic cells, we consider that the analysis of pyroptosis and necroptosis falls outside of the scope of this work and forms a separate independent project which is ongoing in our lab. However, in response to reviewer 3 and this reviewer we now discuss this point in the context of our data on apoptosis (new lines 278-282).

Supplementary Figure 1: Stimulation with PHA + ionomycin is not a physiological activation of human T cells. The authors should also activate primary human T cells with mitogenic antibodies against CD3 and CD28 and then test changes in the surface expression of mucins.

We thank the reviewer for this suggestion. We should point out that we did not use PHA + ionomycin, but as described in the text, PHA + IL-2, and therefore did not include the potentially complicating addition of a calcium ionophore. Whilst we agree that PHA is a less physiologically relevant trigger for T cell activation than CD3 + CD28 stimulation, it is nevertheless a widely used approach to activate T cells in vitro. Moreover, we had already used CD3 + CD28 antibodies presented on beads as an activation stimulus for primary CD4⁺ T cells post-CRISPR-Cas9 to knock out ADAM10, as described in original Supplementary Fig. 4c and in the original methods line 399, and achieved exactly the same mucin phenotype during apoptosis as when we used PHA stimulation (original

Supplementary Fig. 4e). However, we agree that for completeness it is useful to further compare PHA with CD3 + CD28 activation for subsequent apoptosis-induced mucin loss. We have therefore carried out these extra experiments and are pleased to report that we obtain the same pattern of results which are presented in new Supplementary Fig. 2b, and text integrated in new lines 80-81 and 83-85. We have also combined CD3 + CD28 activation of CD4⁺ T cells with glucocorticoid (dexamethasone) induction of apoptosis for maximum physiological relevance and again observe the same pattern of mucin downmodulation (Supplementary Fig. 2i).

Supplementary Figure 2: This experiment should be done also with at least one physiological inducer of apoptosis, such as nutrient deprivation or treatment with glucocorticoids.

Also the experiment should be conducted with stimuli that induce necroptosis or pyroptosis to examine whether the changes in mucin levels on the cell surface are restricted to apoptosis or can also occur when cells undergo different forms of programmed cell death.

NB the reviewer also asks us to repeat these experiments for Figs. 3, 4 and Supplementary Figs. 4, 5

Having demonstrated that CD3/CD28 primary T cell stimulation and dexamethasone-mediated apoptosis induction give exactly the same pattern of mucin downregulation as PHA-stimulated T cells and other well-accepted inducers of apoptosis (staurosporine and UVC), we consider that we have reproduced the central phenomenon under the conditions requested by the reviewer. Since the principal theme of our study is that mucins act as a surface biophysical barrier to efferocytosis rather than (for example) requiring more sophisticated intracellular T cell signalling pathways, we cannot see any merit in reproducing all the other major experiments under the new conditions. Repeating experiments for Figs. 2, 3, 4 and Supplementary Figs. 4, 5 would neither add any greater mechanistic understanding nor any additional conceptual weight to the study, but would delay publication by a very extended period, we estimate by up to a year. Also, as discussed above, we consider that pyroptosis and necroptosis induction are well beyond the scope of the current study which explicitly focusses on apoptosis and efferocytosis, and in agreement with reviewer 3 we do not consider this necessary for the current manuscript.

Figure 4: ADAM10 should be knocked out not only in CEM cells but also in at least one additional T lymphoma cell line to allow generalisation of the findings presented.

We thank the reviewer for this proposal. We remind the reviewer that we had already knocked out ADAM10 in one transformed T cell line (CEM), and in primary CD4⁺ T cells which was technically challenging. Nevertheless in response to the reviewer's request we have been attempting to knock out ADAM10 in Jurkat and HPBALL. Despite using the same guide RNAs as in the previous knock-out experiments we have been unable to isolate viable lines or clones of ADAM10-KO cells. At present we do not know the reason for this but consider that since this further experiment was not requested by the other reviewers, we prefer to resubmit the manuscript in a timely manner with the existing data. We remind the reviewer that we have demonstrated potent activity of the ADAM10-specific inhibitor GI254023X for inhibiting mucin shedding in primary T cells, CEM, Jurkat, HPBALL, NALM6 and U937, and hope that the reviewer will agree with our position.

All Western blots presented need to show molecular weight markers.

We agree and have added molecular weights to the blots, and following Nature publishing guidelines added the entire gels to accompanying Source Data files.

Additional changes to the manuscript

In addition to the new figures and text alluded to above, we have also made the following further changes:

1. We have added flow cytometry gating strategies for mucin analysis of CEM cells and CD4⁺ and CD8⁺ primary T cells (new Supplementary Figs. 2e, f) to clarify how we did the analyses having included CD8⁺ T cells in the revised manuscript, and to comply with Nature data reporting policy. We have also added an explanation of the gating and analysis for primary CD4⁺ and CD8⁺ T cells from monocyte-depleted PBMC to the revised methods section (new lines 315, 321–328, 339, 350-351, 359–360, 373-374).
2. During revision we improved the quality of the blots obtained from supernatant fractions for CD43 and PSGL-1 using exactly the same approach, and have replaced the original blots with these in new Fig. 2f, g, lower blot panels. These cleaner blots allow a clearer representation of the data but in no way change the results or the conclusions of the study. In accordance with Nature data reporting policy we have also added the complete blots for Fig. 2f – h and new Supplementary Figs. 5a, d and 9d, e to the associated Source Data file.
3. We have added new references associated with the new cell lines tested (new refs 22, 47, 48, 50), and the use of actin remodelling inhibitors to block phagocytosis (new refs 34, 35).
4. Panels c and d in main Figure 1 have been swapped over for clarity
5. We have added a new affiliation for the senior author.

REVIEWERS' COMMENTS

Reviewer #1 (Remarks to the Author):

As I mentioned in my first review, the authors demonstrate that apoptotic T cells reduce their “don’t eat me” barrier of mucins (CD43, PSGL-1 and MUC-1) via their shedding by the metalloprotease ADAM10. The latter is shown to be activated via caspase 3-mediated activation of the scramblase XKR8, which flips negatively charged phosphatidylserine from the inner to the outer leaflet of the plasma membrane, which can then engage with a positively charged region on ADAM10 leading to enhanced shedding activity. As a consequence of these events, apoptotic T cell phagocytosis by macrophages (termed efferocytosis) is increased. This is original work that I continue to believe is of high significance and I am happy that the authors have addressed my original comments.

Reviewer #2 (Remarks to the Author):

The authors have adequately responded to my comments and I have no further suggestions. I congratulate the authors on a nice piece of work.

Reviewer #3 (Remarks to the Author):

The authors have adequately addressed all my questions and comments. Thank you very much.

Reviewer #4 (Remarks to the Author):

The authors have performed several of the experiments that I have requested when reviewing the original version of this paper. However, they did not examine whether the reduction in the levels of the mucins of interest is also seen after cells are triggered to undergo necroptosis or pyroptosis, or whether this event is specific to apoptosis. This is a very important question and the experiments to answer these questions are easy to perform. I believe it is not too much to ask the authors to conduct this work.